# Group Distributionally Robust Optimization-Driven RL for LLM Reasoning

**Kishan Panaganti** [1]  **Zhenwen Liang** [2]  **Wenhao Yu** [3]  **Haitao Mi** [1]  **Dong Yu** [4]

## Abstract

Reasoning post-training with GRPO is typically built on *static uniformity*: uniform prompt sampling and a fixed number of rollouts per prompt. For heterogeneous, heavy-tailed reasoning data, this wastes compute on already-solved patterns while under-training the long tail of hard problems. We cast GRPO post-training as **two independent GDRO games** (not coupled) over *dynamic difficulty groups* defined by an *online train-time pass@n* statistic (computed from the rollouts used for GRPO updates): a *data adversary* that reshapes prompt sampling and a *compute adversary* that redistributes rollouts. **Prompt-GDRO** applies multiplicative-weights reweighting over bins (with an EMA-debiased difficulty score) to upweight persistently hard groups without frequency bias. **Rollout-GDRO** allocates rollouts across bins under a fixed *mean* budget via a shadow-price controller, improving gradient information efficiency on high-uncertainty groups while preserving the mean train-time rollout budget. Our approach is principled and theory-driven: we provide no-regret guarantees for the Prompt-GDRO game (via an entropy-regularized GDRO surrogate) and a variance-proxy analysis that yields a square-root optimal compute allocation for Rollout-GDRO. On DAPO 14.1k with Qwen3-Base (1.7B/4B/8B), each controller improves pass@8 by 9–13% over GRPO; targeted Qwen3-4B reruns reproduce the Prompt-GDRO gap under a second seed, show second-seed Rollout-GDRO gains on four of five comparable benchmarks, and show low sensitivity to several key hyperparameters.

[1]Tencent Frontier Lab in Bellevue, WA, USA [2]MiroMind, USA [3]OpenAI, USA [4]Capital One, USA. Correspondence to: Kishan Panaganti <kpb@global.tencent.com>.

*Proceedings of the $43^{rd}$ International Conference on Machine Learning*, Seoul, South Korea. PMLR 306, 2026. Copyright 2026 by the author(s).

## 1. Introduction

The capabilities of Large Language Models (LLMs) in complex reasoning are increasingly shaped not only by architectural scaling, but by the design of post-training objectives and alignment pipelines.[0] Reinforcement Learning (RL), particularly methods like Proximal Policy Optimization (PPO) (Schulman et al., 2017) and Group Relative Policy Optimization (GRPO) (Shao et al., 2024), has emerged as a standard approach for aligning models with rigorous logical constraints. By optimizing for sparse verifiable rewards or process-based supervision, these methods have enabled significant breakthroughs in mathematical problem solving (Wang et al., 2024b) and code generation.

At a high level, our perspective is that reasoning post-training exposes *two* distinct sources of non-uniformity: (i) *which prompts* remain unsolved as the model improves (a shifting difficulty landscape), and (ii) *how much exploration* different prompts require to yield low-variance learning signals. Standard pipelines treat both knobs as static—uniform prompt sampling and fixed rollouts—which we argue is mismatched to the heavy-tailed structure of reasoning.

Recent discussions in the broader ML community further motivate moving beyond uniformity from a *data value* perspective. Finzi et al. (Finzi et al., 2026) propose *epiplexity*—a notion of information that aims to quantify the structural content that a computationally bounded learner can extract from data (distinct from time-bounded entropy/noise). From this viewpoint, the "value" of a prompt is not determined by its frequency, but by whether it still contains learnable structure *for the current policy* under a fixed compute budget. This perspective is aligned with our Prompt-GDRO mechanism: by steering sampling toward the evolving reasoning frontier, we concentrate updates on prompts that remain high-value for learning, rather than repeatedly training on already-solved, low-value examples.

Concurrently, empirical analyses of RL with verifiable rewards suggest that learning progress can be dominated by a small fraction of high-entropy "decision" points, challenging the implicit assumption that uniform averaging over all tokens/prompts is the most compute-efficient choice (Wang

---

[0]Adam Marblestone (paraphrased) on Dwarkesh's podcast (YouTube): "The brain's secret sauce is its loss functions, not its architecture."

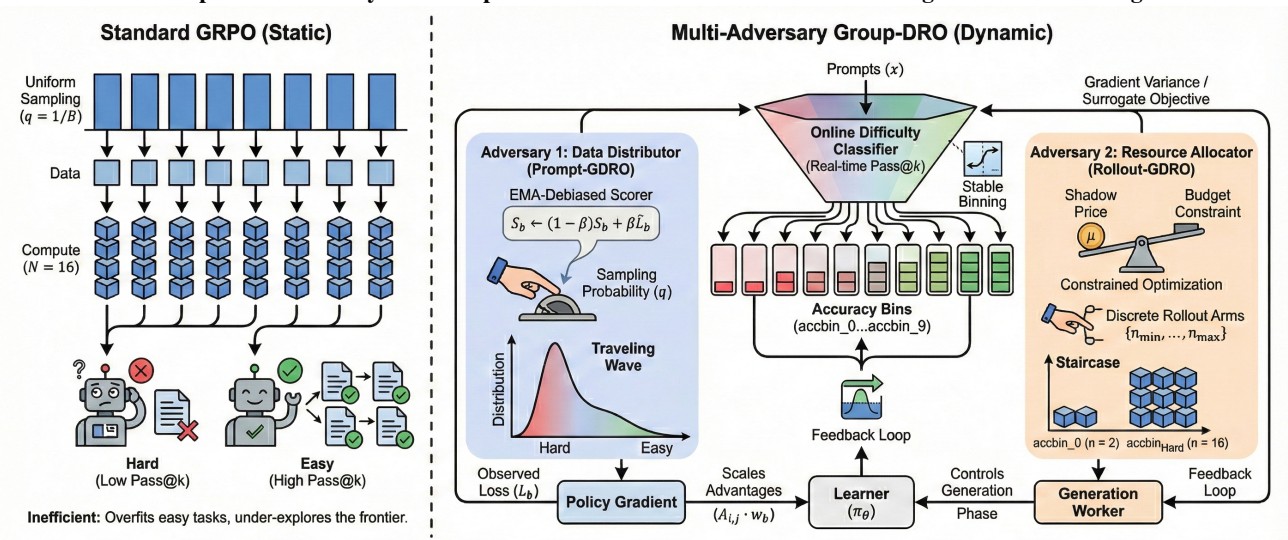

**Figure 1. Conceptual Illustration: Static Uniformity vs. Multi-Adversary GDRO (Dynamic).** (Left) Standard GRPO samples prompts uniformly ($q = 1/B$) and assigns a fixed number of rollouts (schematically $N = 16$), causing it to overfit easy tasks while under-exploring the frontier. (Right) Our framework employs an **Online Difficulty Classifier** to dynamically partition prompts based on an online train-time pass@$n$ estimate (any-of-$n$ success computed from the rollouts generated for GRPO updates). It introduces two *independent* adversarial feedback loops (not coupled): (1) **Prompt-GDRO** (Data Distributor) uses an EMA-debiased scorer to shift the sampling distribution toward hard bins, creating a "traveling wave" of difficulty; (2) **Rollout-GDRO** (Resource Allocator) uses a shadow price $\mu$ to solve a constrained optimization problem, allocating discrete rollout arms ($n_{\min} \ldots n_{\max}$) to maximize gradient variance reduction on high-uncertainty tasks.

et al., 2025). Related work on "thinking" and the accuracy–compute Pareto frontier emphasizes that additional computation is most beneficial when applied selectively, and can be inefficient when applied uniformly across instances (Madaan et al., 2025). Taken together, these perspectives echo a community intuition that *both* data selection and compute allocation should be adaptive (Finzi, 2026)—precisely the two control knobs instantiated by our Prompt-GDRO and Rollout-GDRO adversaries.

However, prevalent RL pipelines rely on a fundamental assumption of *static uniformity*: they sample prompts uniformly from the training distribution and allocate a fixed computational budget (number of rollouts) to every prompt. We argue that this rigidity creates structural inefficiencies. As formalized in our analysis (Section 3), reasoning datasets are inherently heterogeneous, composed of disjoint sub-domains (e.g., elementary algebra vs. Olympiad number theory) with vastly different difficulty profiles. Under uniform sampling, optimization is dominated by the most frequent, often easier patterns, so learning signal concentrates on the "easy core" while errors persist in a long, difficult tail. A long line of supervised learning work has addressed this asymmetry by making training explicitly *difficulty-aware*—from curriculum and self-paced learning that schedule examples from easy to hard (Bengio et al., 2009; Kumar et al., 2010), to boosting and hard-example mining that upweight misclassified or high-loss instances (Freund & Schapire, 1997; Shrivastava et al., 2016), and focal losses that downweight well-classified examples (Lin et al., 2017). This motivates viewing robustness through *difficulty-defined* groups and

optimizing worst-bin performance in the spirit of GDRO (Sagawa et al., 2020). Furthermore, the value of computational exploration is non-uniform. In reasoning tasks, "solved" prompts yield low-variance gradients, while high-entropy "frontier" prompts require massive exploration to reduce gradient variance (Setlur et al., 2025). A static budget allocation fails to capture this dynamic, wasting resources on redundant verification while under-exploring critical failure modes.

To address these limitations, we propose **Multi-Adversary Group Distributionally Robust Optimization (GDRO)** framework. Motivated by the biological hypothesis that intelligent systems distinguish between a core representation learning module and a specialized "steering subsystem" that optimizes cost functions (Marblestone et al., 2016), we implement a dynamic, data-agnostic grouping mechanism. Specifically, we replace static uniformity with an *Online Difficulty Classifier* that partitions data based on an online train-time pass@$n$ statistic computed from the rollouts generated for GRPO updates, effectively allowing the optimization process to "steer" itself. We then formulate post-training as a zero-sum game and instantiate two complementary adversaries via the **GDRO-EXP3P** algorithm (Soma et al., 2022). Importantly, *these adversaries are designed as independent modules*: Prompt-GDRO plays a GDRO reweighting game against the learner, while Rollout-GDRO plays a separate constrained compute-allocation game. In this work we analyze and evaluate the two games in isolation (no coupling), i.e., all experiments enable Prompt-GDRO or Rollout-GDRO individually; jointly coupling both adver-

saries into a single multi-time-scale system is left to future work.

Our contributions are summarized as follows:

1. **Prompt-GDRO (A Data Adversary):** We employ an adversarial reweighting rule that targets the *intensive* difficulty margin (mean loss) instead of uniform sampling. We introduce an *EMA-Debiased* scoring rule to prevent the adversary from succumbing to frequency bias, ensuring that rare, high-difficulty groups are upweighted effectively. This acts as a regularizer against over-optimizing the easy core, improving worst-bin robustness as the difficulty frontier shifts. *Theory:* In Section 3 (proofs in Appendix B), we show that exponential-weights Prompt-GDRO corresponds to optimizing an entropy-regularized GDRO surrogate (a log-sum-exp "soft worst-group" objective) and admits a no-regret game interpretation.

2. **Rollout-GDRO (A Compute Adversary):** We challenge the convention of fixed rollout budgets (e.g., $n = 4$). We formulate rollout allocation as a constrained resource allocation game where a second adversary dynamically assigns rollout counts to maximize gradient variance reduction on hard tasks, subject to a global *mean-rollout* budget constraint. This enables the model to allocate deeper exploration to difficult problems without increasing the mean number of train-time rollouts. *Theory:* In Section 3 (proofs in Appendix B), we derive a variance proxy for GRPO rollouts and show that the variance-optimal mean-budget allocation obeys a square-root law, motivating the shadow-price controller used by Rollout-GDRO.

3. **Empirical results.** On the DAPO 14.1k reasoning dataset with Qwen3-Base models (1.7B/4B/8B), Prompt-GDRO improves pass@8 by **+9.74%**, **+13.13%**, and **+8.96%**, and Rollout-GDRO by **+10.64%**, **+10.59%**, and **+9.20%** over GRPO. Additional Qwen3-4B checks reproduce the Prompt-GDRO gap under a second seed, show Rollout-GDRO seed-2 gains on four of five comparable benchmarks, and test one-at-a-time sensitivity to bin count, history length, adversary learning rate, and weight cap. Qualitative analyses reveal an emergent curriculum that shifts sampling weight and rollout budget toward the evolving reasoning frontier.

## 2. Preliminaries

### 2.1. Reinforcement Learning for Reasoning

We formalize post-training for reasoning as Reinforcement Learning (RL) over an autoregressive language policy. Let $x \sim \mathcal{D}$ denote a prompt and let $y = (y_1, \ldots, y_L)$ denote a response sampled from a policy $\pi_\theta(\cdot \mid x)$ parameterized by

$\theta$. The conditional sequence probability factorizes as

$$\pi_\theta(y \mid x) = \prod_{t=1}^{L} \pi_\theta(y_t \mid x, y_{<t}). \qquad (1)$$

The RL objective is to maximize expected reward

$$J(\theta) = \mathbb{E}_{x \sim \mathcal{D}, \, y \sim \pi_\theta(\cdot \mid x)}[r(x, y)], \qquad (2)$$

where $r(x, y)$ is a task-dependent, generally non-differentiable reward. In mathematical reasoning, $r(x, y)$ is typically sparse (e.g., binary correctness) or semi-sparse (e.g., verifier-based signals).

### 2.2. Group-Relative Policy Optimization (GRPO)

Group-Relative Policy Optimization (GRPO) (Shao et al., 2024) is a computationally efficient alternative to Proximal Policy Optimization (PPO) (Schulman et al., 2017). GRPO eliminates a learned value critic by constructing a baseline from a within-prompt group of rollouts.

*Remark* 2.1 (Two notions of "group"). Throughout the paper, the term "group" can refer to two different objects: (i) a *GRPO rollout group* (multiple rollouts for a fixed prompt), and (ii) a *GDRO group/bin* (a subset of prompts induced by a grouping rule, e.g., an online difficulty bin). We explicitly index GRPO rollouts by $(i, j)$ and GDRO bins by $b \in \{1, \ldots, B\}$.

*Remark* 2.2 (Notation: $n$ vs. $k$). We use $n$ for the GRPO rollout-group size (train-time rollouts per prompt), and $k$ for "best-of-$k$" *evaluation* statistics such as pass@k and mean@k. Our *online difficulty classifier* uses a train-time pass@$n$ statistic computed from the rollouts actually generated for GRPO updates (so $n = 4$ for Prompt-GDRO; under Rollout-GDRO, $n$ can be bin-dependent). In our experiments, the base training rollout group size is $n = 4$ and evaluation uses $k = 8$.

**Group sampling and advantage estimation.** For each prompt $x_i$, we sample $n$ responses $\{y_{i,j}\}_{j=1}^{n}$ from a behavior policy $\pi_{\theta_{\text{old}}}$ (the policy used to generate rollouts). Each response receives a scalar reward $r_{i,j} \triangleq r(x_i, y_{i,j})$. GRPO computes a group-relative advantage by standardizing rewards within the rollout group:

$$A_{i,j} = \frac{r_{i,j} - \mu_i}{\sigma_i + \varepsilon}, \qquad (3)$$

where $\mu_i \triangleq \frac{1}{n} \sum_{j=1}^{n} r_{i,j}$, $\sigma_i \triangleq \sqrt{\frac{1}{n} \sum_{j=1}^{n} (r_{i,j} - \mu_i)^2}$, and $\varepsilon > 0$ is a small constant for numerical stability. The scalar advantage $A_{i,j}$ is applied token-wise to all tokens in $y_{i,j}$ in the surrogate objective below.

**The clipped surrogate objective.** Let

$$\rho_{i,j,t}(\theta) \triangleq \frac{\pi_\theta(y_{i,j,t} \mid x_i, y_{i,j,<t})}{\pi_{\theta_{\text{old}}}(y_{i,j,t} \mid x_i, y_{i,j,<t})} \qquad (4)$$

denote the token-level importance ratio. The GRPO/PPO-style clipped surrogate for response $y_{i,j}$ is $\mathcal{J}_{i,j}^{\mathrm{CLIP}}(\theta) =$

$$\frac{1}{L_{i,j}} \sum_{t=1}^{L_{i,j}} \min\Big( \rho_{i,j,t}(\theta)\, A_{i,j},\; \mathrm{clip}\big(\rho_{i,j,t}(\theta),\, 1-\epsilon,\, 1+\epsilon\big)\, A_{i,j}\Big),$$
(5)

where $\epsilon > 0$ is the PPO clipping parameter.

**Observable loss signal.** For our robust optimization controllers, we require a scalar "loss-like" signal per generated response. We define the per-response loss as the negative KL-regularized surrogate:

$$\ell_{i,j}(\theta) \;=\; -\mathcal{J}_{i,j}^{\mathrm{CLIP}}(\theta) + \beta_{\mathrm{KL}}\, D_{\mathrm{KL}}(\pi_\theta(\cdot \mid x_i) \,\|\, \pi_{\mathrm{ref}}(\cdot \mid x_i)),$$
(6)

where $\beta_{\mathrm{KL}} > 0$ controls the KL penalty to a fixed reference policy $\pi_{\mathrm{ref}}$. In our experiments we operate in a *zero-SFT* setting, so $\pi_{\mathrm{ref}}$ is simply the *initial base checkpoint* (i.e., the same Qwen3-{1.7B,4B,8B}-Base model we start RL from) and is held frozen during training. In practice, $D_{\mathrm{KL}}(\pi_\theta(\cdot \mid x_i) \,\|\, \pi_{\mathrm{ref}}(\cdot \mid x_i))$ is evaluated on the sampled responses using token-level log-probabilities under $\pi_\theta$ and $\pi_{\mathrm{ref}}$. Finally, we define the *prompt-level* loss as the mean over rollouts:

$$\ell(x_i;\theta) \;\triangleq\; \frac{1}{n} \sum_{j=1}^{n} \ell_{i,j}(\theta).$$
(7)

This prompt-level signal will be aggregated by bins and fed to the GDRO adversaries.

### 2.3. Group Distributionally Robust Optimization

Standard Empirical Risk Minimization (ERM) minimizes average loss under the empirical mixture, which can be dominated by high-frequency and/or easy instances. A long line of supervised learning work addresses this imbalance by making training explicitly *difficulty-aware*—from curriculum and self-paced learning that schedule examples from easy to hard (Bengio et al., 2009; Kumar et al., 2010), to boosting and hard-example mining that upweight misclassified or high-loss instances (Freund & Schapire, 1997; Shrivastava et al., 2016), and focal losses that downweight well-classified examples (Lin et al., 2017). GDRO provides a complementary, principled objective: treat subpopulations (here, difficulty bins) as groups and optimize worst-group risk.

We adopt **Group Distributionally Robust Optimization** (GDRO) (Sagawa et al., 2020). Assume prompts are partitioned into $B$ disjoint groups (domains) $\{\mathcal{D}_1, \ldots, \mathcal{D}_B\}$. Let

$$L_b(\theta) \;\triangleq\; \mathbb{E}_{x \sim \mathcal{D}_b}[\ell(x;\theta)]$$
(8)

denote the expected prompt-level loss for group $b$. GDRO

optimizes worst-group performance by solving

$$\min_\theta\; \max_{q \in \Delta_B}\; \sum_{b=1}^{B} q_b\, L_b(\theta),$$
(9)

where $\Delta_B$ is the probability simplex. Following Soma et al. (2022), (9) admits a zero-sum game interpretation between a learner ($\theta$) and an adversary ($q$) and can be optimized via no-regret online learning. Our method instantiates this perspective with multiple adversarial "levers" on top of the GRPO loss signal in (6).

## 3. Multi-Adversary GDRO Formulation

We analyze why *static uniformity*—uniform prompt sampling and a fixed rollout budget per prompt—can be structurally inefficient for reasoning post-training, and how our two adversaries implement targeted robustness improvements. Our main message is that (i) **Prompt-GDRO** realizes an entropy-regularized GDRO objective over *online difficulty groups*, and (ii) **Rollout-GDRO** realizes a *mean-rollout-budget preserving* variance-aware rollout allocation. Proofs and extended technical discussion are deferred to Appendix E.

### 3.1. Setup: Online difficulty groups and two adversarial levers

Reasoning datasets are heterogeneous: prompts belong to latent sub-domains with widely varying difficulty. Once a large fraction of prompts become "nearly solved," their rollouts yield low-variance gradients and diminishing marginal learning signal, while the remaining frontier prompts continue to exhibit high uncertainty. Uniform averaging can therefore over-optimize the "easy core" and under-train the long tail (Bengio et al., 2009; Sagawa et al., 2020).

**Online difficulty bins.** Let $\ell(x;\theta)$ denote the GRPO prompt-level loss signal from Section 2.2 (Eq. (6)). For each prompt $x$, we track an online success statistic using an *any-of-the-generated* correctness indicator. Let $n_t(x)$ denote the number of rollouts actually generated for $x$ at step $t$ (e.g., $n_t(x) \equiv n$ for GRPO; under Rollout-GDRO, $n_t(x)$ is bin-dependent). We define

$$c_t(x) \triangleq \mathbb{1}\{\exists j \in \{1, \ldots, n_t(x)\} : r(x, y_j) = 1\}, \quad (10)$$

and maintain a length-$H$ sliding-window estimate

$$\widehat{\mathrm{pass}}_t(x) \;\triangleq\; \frac{1}{H} \sum_{s=t-H+1}^{t} c_s(x).$$
(11)

In words, $\widehat{\mathrm{pass}}_t(x)$ estimates the recent probability that *at least one* sampled rollout solves $x$ under the train-time rollout budget. This statistic is computed from the same rollouts

used for GRPO updates, so it introduces no additional sampling overhead; separately, we report *evaluation* pass@8 in Table 1. Given bin edges $0 = a_0 < a_1 < \cdots < a_B = 1$, we assign $x$ to bin $b = g_t(x)$ if $\widehat{\text{pass}}_t(x) \in [a_{b-1}, a_b)$, inducing time-varying groups $\{\mathcal{D}_{t,b}\}_{b=1}^B$. Within a step $t$, treat $g_t$ as fixed and define bin-conditional risks

$$L_{t,b}(\theta) \triangleq \mathbb{E}[\ell(x;\theta) \mid g_t(x) = b], \qquad b \in \{1,\ldots,B\}. \tag{12}$$

Implementation details and hyperparameters for the online binning are summarized in Appendix C.

**A multi-adversary robust objective.** With bins fixed, GDRO optimizes worst-bin loss via

$$\min_\theta \ \max_{q \in \Delta_B} \ \sum_{b=1}^B q(b) \, L_{t,b}(\theta). \tag{13}$$

We instantiate *two* adversarial "control knobs" on top of GRPO:

1. **Prompt-GDRO** updates an adversarial bin distribution $q_t \in \Delta_B$ and applies it without changing the number of rollouts by reweighting GRPO updates from prompts in each bin.
2. **Rollout-GDRO** chooses bin-specific rollout counts $n_b(t) \in \{n_{\min}, \ldots, n_{\max}\}$ under a strict mean-rollout constraint $\sum_b \hat{q}_t(b) \, n_b(t) = \bar{n}$ (where $\hat{q}_t$ is the realized bin share in the batch), reallocating compute to improve gradient signal-to-noise.

### 3.2. Prompt-GDRO: Entropy-regularized GDRO via exponential weights

Prompt-GDRO implements the inner adversary in Eq. (13) using exponential weights. A convenient interpretation is through an entropy-regularized "soft worst-bin" surrogate, which yields a no-regret game interpretation of the learner–adversary dynamics (Appendix E.1.2).

**Entropic surrogate and softmax weights.** For $\eta > 0$, define the entropy-regularized inner problem

$$\mathcal{R}_\eta(\theta) \triangleq \max_{q \in \Delta_B} \left\{ \sum_{b=1}^B q(b) L_{t,b}(\theta) + \frac{1}{\eta} H(q) \right\}, \tag{14}$$

where $H(q) \triangleq -\sum_{b=1}^B q(b) \log q(b)$. This has the closed form

$$\mathcal{R}_\eta(\theta) = \frac{1}{\eta} \log \left( \sum_{b=1}^B e^{\eta L_{t,b}(\theta)} \right), \tag{15}$$

with unique maximizer

$$q_\eta(b;\theta) = \frac{\exp(\eta L_{t,b}(\theta))}{\sum_{j=1}^B \exp(\eta L_{t,j}(\theta))}. \tag{16}$$

Moreover, $\mathcal{R}_\eta$ approximates the hard worst-bin loss up to $\log B/\eta$: $\max_b L_{t,b}(\theta) \leq \mathcal{R}_\eta(\theta) \leq \max_b L_{t,b}(\theta) + \log(B)/\eta$. Differentiating (15) yields the mixture-gradient form $\nabla_\theta \mathcal{R}_\eta(\theta) = \sum_b q_\eta(b;\theta) \nabla_\theta L_{t,b}(\theta)$.

**Online, frequency-robust reweighting.** In post-training, bin losses are noisy and bins evolve with the policy. Prompt-GDRO maintains a smoothed *intensive* difficulty estimate per bin (an EMA of mean prompt losses) and uses an EXP3P-style update to track a softmax best response under bandit noise. To mitigate frequency bias, the score can be debiased by the realized bin share so that rare but persistently hard bins remain competitive (see Appendix E.1.2).

**Budget-preserving implementation.** Prompt-GDRO realizes adversarial pressure via per-bin multipliers on GRPO advantages, without changing the rollout count. Let $\omega_t(b)$ denote the EXP3-style weight for bin $b$ at step $t$; the practical counterpart of (16) with EMA-debiased loss. For a prompt $x_i$ in bin $b = g_t(x_i)$, we scale its rollout advantages and apply a stability clip:

$$A_{i,j} \ \leftarrow \ A_{i,j} \cdot \min\{\omega_t(b), \omega_{\max}\}. \tag{17}$$

Intuitively, this "pays" more gradient budget to hard bins while preserving the train-time rollout count.

### 3.3. Rollout-GDRO: Variance-aware compute allocation under a mean-budget constraint

Rollout-GDRO reallocates rollouts across bins while keeping the *mean* rollout count fixed, targeting bins where extra exploration most improves the training signal.

**Mean-budget rollout objective.** Let $n_b(t)$ denote the rollouts assigned to prompts in bin $b$ at step $t$, and let $\hat{q}_t(b)$ be the realized bin share. Rollout-GDRO enforces the strict mean-budget constraint $\sum_{b=1}^B \hat{q}_t(b) n_b(t) = \bar{n}$ and chooses $n_b(t)$ to be adversarially informative. A generic formulation is a constrained best response

$$\max_{\{n_b\}} \ \sum_{b=1}^B \hat{q}_t(b) \, \widehat{J}_{t,b}(\theta; n_b) \quad \text{s.t.} \quad \sum_{b=1}^B \hat{q}_t(b) \, n_b = \bar{n}, \tag{18}$$

where $\widehat{J}_{t,b}$ is a bin-level utility computed from the rollouts actually taken with arm $n_b$. In our implementation, $\widehat{J}_{t,b}(\theta; n_b)$ is the negative mean GRPO loss in bin $b$ (equivalently, the mean clipped-policy-gradient objective), estimated from those rollouts. The discrete-arm EXP3P+shadow-price controller we use is a no-regret algorithm for solving (18) online (Appendix E.2.4).

**A variance proxy.** (18) describes the *implemented* rollout-allocation problem. Separately, to explain why reallocating

rollouts improves optimization, we analyze how $\mathbf{n}$ affects the *stochasticity* of GRPO's Monte Carlo gradients. Under i.i.d. rollout sampling and a mild bounded-differences condition on the prompt-level GRPO gradient estimator (which covers within-prompt normalization across the rollout group), using $n_b$ rollouts for prompts in bin $b$ yields a canonical $1/n_b$ reduction in the conditional variance of the per-prompt gradient noise (Appendix E.2.1). Abstractly, let $v_b(\theta)$ denote a bin-dependent intrinsic variance parameter. A simple proxy for batch-level noise is

$$\text{VarProxy}(\mathbf{n}; \theta) \triangleq \sum_{b=1}^{B} \hat{q}_t(b) \frac{v_b(\theta)}{n_b}, \quad (19)$$

where $\mathbf{n} = (n_1, \ldots, n_B)$ and $\hat{q}_t$ is the realized bin share in the batch. Appendix E.2.1 derives this proxy under a concrete sampling model and connects it to GRPO's Monte Carlo estimators. When the rollout controller is instantiated with a utility that (directly or via a monotone transform) estimates *negative* variance proxy cost, the same primal–dual no-regret analysis yields an additional guarantee: the learned allocation is provably near-optimal for the variance-minimization relaxation (20), and therefore minimizes (19) up to sublinear regret (Appendix E.2.4).

Consider the continuous relaxation of mean-budget variance minimization:

$$\min_{\mathbf{n} \in \mathbb{R}_+^B} \sum_{b=1}^{B} \hat{q}_t(b) \frac{v_b(\theta)}{n_b} \quad \text{s.t.} \quad \sum_{b=1}^{B} \hat{q}_t(b) n_b = \bar{n}. \quad (20)$$

The unique optimum on active bins satisfies the Neyman/square-root rule

$$n_b^\star = \bar{n} \cdot \frac{\sqrt{v_b(\theta)}}{\sum_{j=1}^{B} \hat{q}_t(j) \sqrt{v_j(\theta)}} \quad \propto \quad \sqrt{v_b(\theta)}, \quad (21)$$

with proof in Appendix E.2.2. Equivalently, the KKT condition admits a shadow-price interpretation: for $\mu > 0$, the per-bin best response to the Lagrangian $\frac{v_b(\theta)}{n_b} + \mu n_b$ is $n_b(\mu) = \sqrt{v_b(\theta)/\mu}$, and $\mu$ is chosen to satisfy the mean-budget constraint.

**Discrete arms and online control.** In practice, Rollout-GDRO chooses $n_b(t)$ from a discrete set of rollout arms and only observes bandit feedback for the chosen arm. We implement the controller via EXP3P updates over arms together with a dual ascent update for the shadow price $\mu$ (Appendix E.2.3). This closed-loop control is what produces the "budget frontier" patterns and the variance-reduction diagnostic in Figure 6b.

# 4. Experiments

In this section, we present the empirical evaluation of our **Multi-Adversary GDRO** framework. We use the *14.1k*

*English subset* of the DAPO-Math-17k-Processed dataset[1] (we refer to this training set as *DAPO-14.1k*) for all training runs, following a standard post-training pipeline.

**Metrics.** Unless stated otherwise, we report *mean@8* for benchmark accuracies: each prompt is evaluated with 8 sampled completions and we average the binary correctness indicator across those 8 trials. We additionally report *pass@8*, the probability that at least one of the 8 completions is correct (estimated from the same samples). This distinction matters for reasoning: mean@8 reflects typical performance, while pass@8 captures "best-of-$k$" robustness under limited search.

**Rollout-budget neutrality.** Prompt-GDRO keeps the rollout budget fixed and changes training pressure through binwise reweighting of GRPO updates. Rollout-GDRO keeps the *mean* rollout budget fixed (e.g., $\bar{n} = 4$) and redistributes rollouts across bins. Thus, improvements reflect better use of the same mean train-time rollout budget rather than a larger sampling budget. This is a rollout-budget statement, not an elapsed-time-neutrality claim; we discuss measured overhead and targeted Qwen3-4B second-seed checks below.

We first report the quantitative improvements on standard mathematical reasoning benchmarks. Subsequently, we provide a rigorous qualitative analysis of the dynamic difficulty landscape, interpreting how the interaction between model capacity and data heterogeneity drives the emergent curriculum observed in our prompt reweighting distribution and rollout allocation strategies.

## 4.1. Main Results

We evaluate our method across three model scales: *Qwen3-1.7B-Base*, *Qwen3-4B-Base*, and *Qwen3-8B-Base*. We compare the standard GRPO baseline against our two proposed mechanisms: *Prompt-GDRO* (adversarial prompt reweighting driven by online difficulty bins) and *Rollout-GDRO* (adversarial compute budgeting). Table 1 summarizes the performance at the peak checkpoint for each stabilized run.

Our framework demonstrates consistent gains across these settings. **Prompt-GDRO** consistently improves performance by explicitly targeting hard data groups, achieving a peak gain of **+13.13%** on the 4B model. Remarkably, **Rollout-GDRO** achieves comparable or superior results—improving the 1.7B model by **+10.64%** and the 8B model by **+9.20%**—without altering the data distribution. This supports our hypothesis that *compute allocation* is a powerful lever for robustness: by dynamically assigning more rollouts to high-variance prompts, the adversary reduces gradient variance where the model is most uncertain, yielding

---

[1] https://huggingface.co/datasets/open-r1/DAPO-Math-17k-Processed

*Table 1.* Results on mathematical reasoning benchmarks for Prompt Reweighting GDRO and Rollout Budgeting GDRO vs GRPO Baseline. **Bold** values indicate methods that outperform the GRPO Baseline (independent comparison). The **AIME** column reports the average accuracy of AIME 2024 and AIME 2025. The percentage improvement for pass@8 is shown in brackets. All other metrics reported are **mean@8**.

| Models | MATH 500 | AIME | AMC | MINERVA | OLYMPIAD | GPQA | pass@8 |
|---|---|---|---|---|---|---|---|
| *Qwen3-1.7B-Base* | | | | | | | |
| GRPO (Baseline) | 50.62 | 5.42 | 34.69 | 14.56 | 23.07 | 26.39 | 50.74 |
| Prompt-GDRO | **63.20** | **6.88** | **39.38** | 14.61 | **25.07** | **29.29** | **55.68** (+9.74%) |
| Rollout-GDRO | **63.98** | **7.50** | **36.87** | **17.28** | **26.71** | 27.72 | **56.14** (+10.64%) |
| *Qwen3-4B-Base* | | | | | | | |
| GRPO (Baseline) | 72.05 | 11.25 | 60.94 | 17.79 | 30.48 | 35.54 | 56.31 |
| Prompt-GDRO | **75.78** | **12.92** | **64.06** | **26.72** | **40.98** | **40.88** | **63.70** (+13.13%) |
| Rollout-GDRO | **75.20** | **13.96** | **67.50** | **26.47** | **39.28** | **38.51** | **62.27** (+10.59%) |
| *Qwen3-8B-Base* | | | | | | | |
| GRPO (Baseline) | 73.45 | 14.38 | 66.56 | 28.17 | 36.92 | 42.25 | 62.04 |
| Prompt-GDRO | **76.18** | **16.04** | **70.94** | **32.17** | **42.43** | **43.81** | **67.60** (+8.96%) |
| Rollout-GDRO | **77.88** | **15.63** | **66.88** | **29.55** | **43.62** | **43.31** | **67.75** (+9.20%) |

gains that can rival direct data curriculum learning.

---

**Key Finding 1**

Both prompt reweighting (Prompt-GDRO) and adaptive compute allocation (Rollout-GDRO) independently yield significant performance gains, improving pass@8 accuracy by up to 13.13% and 10.64% respectively across model scales.

---

### 4.2. Qualitative Analysis: The Dynamics of Difficulty (Prompt-GDRO)

To understand the mechanism behind these performance gains, we visualize the temporal evolution of the training distribution. Figure 2 presents a comprehensive triptych tracking the causal chain of our adversarial mechanism: from *data availability* (prompt share) to *adversarial pressure* (weights) to *learning payoff* (reward).

**Additional qualitative analysis.** We defer additional curriculum diagnostics to Appendix D.

---

**Key Finding 2**

Prompt-GDRO generates an emergent curriculum that decouples the learning signal from dataset frequency. It applies disproportionate pressure to rare, hard bins, creating a "traveling wave" of difficulty that adapts to the model's capacity.

---

### 4.3. Qualitative Analysis: Adaptive Compute Allocation (Rollout-GDRO)

While Prompt-GDRO improves robustness by altering *what* data the model sees, Rollout-GDRO improves robustness by altering *how deeply* the model explores that data. To under-

stand this mechanism, we analyze the adversarial budgeter's behavior through three complementary visualizations: the continuous budget frontier, macro-level scalar dynamics, and discrete allocation snapshots.

**The Budget Frontier** Figure 3 visualizes the direct contrast between data frequency and resource allocation. The left column shows the prompt share (dataset mass), while the right column shows the allocated rollout budget per prompt.

This comparison offers direct qualitative evidence for our method's economic policy. For the 8B model (bottom row), the dataset mass (left) remains concentrated in easier bins for hundreds of steps. However, the budgeter (right) rapidly identifies the emerging capability in accbin_6 and above, locking high-compute resources onto these rare prompts. In contrast, the 1.7B model (top row) takes significantly longer to leave accbin_0, and the budget frontier moves more slowly, consistent with a capacity-dependent curriculum pace.

**Macro-Dynamics of Allocation** To quantify these trends, we provide macro-level training dynamics, variance-aware efficiency diagnostics, and discrete allocation snapshots in Appendix D.

**Discrete Economic Phases** We provide discrete allocation snapshots in Appendix D, illustrating the "multiplier effect" where rare, high-value prompts receive substantially more rollouts than under a uniform baseline.

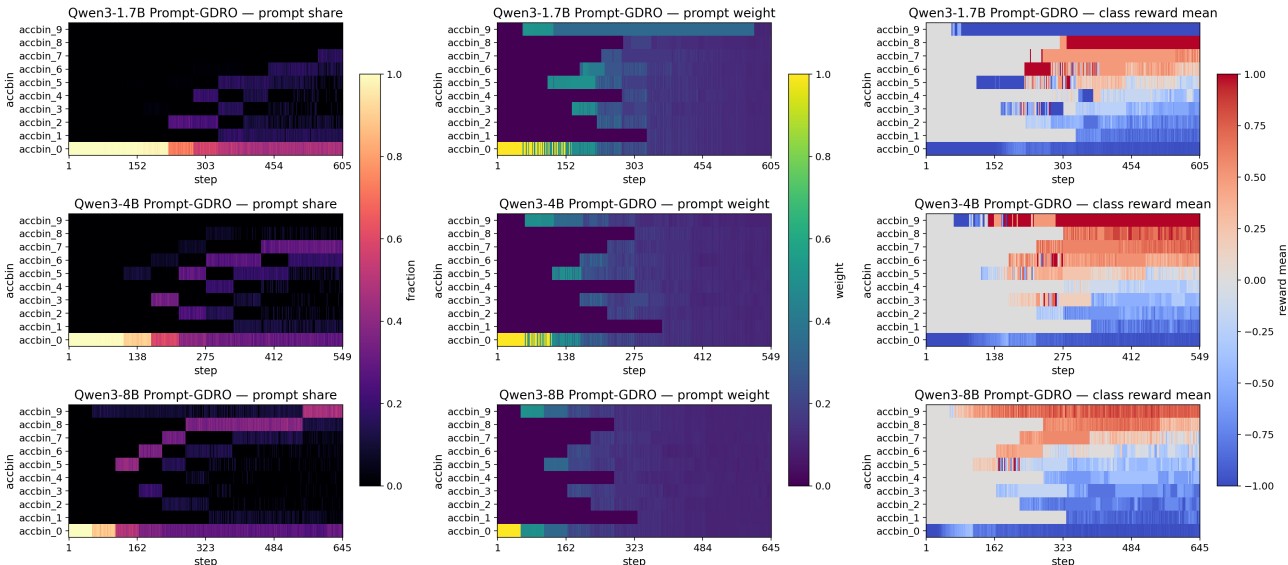

*Figure 2.* **Curriculum Chain.** A triptych comparing the **Prompt Distribution** (Left), **Adversarial Weights** (Middle), and **Realized Reward** (Right; per-bin mean of the scalar verifier reward $r(x, y) \in [-1, 1]$ over rollouts generated at each step) for 1.7B, 4B, and 8B models. This visualizes the mechanism: even when hard bins are rare in the data (dark regions in Left), the adversary applies disproportionate pressure (bright bands in Middle), encouraging learning on these problems and eventually yielding positive rewards (emergence of red/blue bands in Right). The pattern supports that Prompt-GDRO decouples the learning signal from dataset frequency.

---

### Key Finding 3

Rollout-GDRO autonomously identifies the "transition zone" of difficulty and puts computational resources there. This results in a highly non-uniform economic policy where rare, high-value prompts receive up to $10\times$ more rollouts than a uniform baseline, improving gradient information efficiency.

---

### 4.4. Additional Robustness and Sensitivity Checks

We add targeted checks for three practical questions: seed robustness, hyperparameter brittleness, and whether simpler rollout heuristics can explain the gains. Table 2 summarizes the Qwen3-4B checks added for the camera-ready version, including second-seed evidence for both Prompt-GDRO and Rollout-GDRO.

*Table 2.* Targeted Qwen3-4B checks. Numbers are overall micro-average pass@8 unless noted.

| Variant | Step | Result |
|---|---|---|
| Prompt-GDRO, seed 2 | 200 | 63.47 |
| Prompt-GDRO, 10 bins | 200 | 64.11 |
| Prompt-GDRO, $H = 50$ | 200 | 63.70 |
| Prompt-GDRO, $\eta_q = 0.30$ | 200 | 63.99 |
| Prompt-GDRO, cap $= 5$ | 200 | 62.39 |
| Rollout-GDRO, seed 2 | 200 | 4/5 bench gains |
| Rollout-GDRO, 6-bin variant | 150 | 56.16 |
| Rollout-GDRO, arms $[2, 8]$ | 250 | 56.12 |
| Rollout-GDRO, dual LR $= 0.01$ | 150 | 55.35 |
| StaticRollout-Hard | 200 | 56.70 |
| StaticRollout-Easy | 400 | 54.34 |

**Prompt-GDRO seed and sensitivity.** The second-seed Prompt-GDRO run reproduces the main 4B gap: 63.47 overall pass@8 versus 56.36 for matched GRPO, compared with 63.70 versus 56.31 in the original 4B runs. The four one-at-a-time variants stay within $[-1.31, +0.41]$ pp of the paper configuration, and all remain at least $+6$ pp above the GRPO baseline. The near-matched absolute scores and relative gap across seeds support the Prompt-GDRO robustness conclusion rather than a favorable-seed or knife-edge-hyperparameter explanation.

**Online classifier stability.** The difficulty signal itself changes smoothly rather than oscillating wildly. Across the original 1.7B/4B/8B Prompt-GDRO traces, the mean one-step $\ell_1$ movement of bin-share mass is 0.070/0.079/0.086 and the mean one-step KL is 0.011/0.013/0.013, while the first-to-last $\ell_1$ drift is 1.30/1.55/1.50. Thus the classifier is not frozen—it tracks substantial migration of prompts across difficulty bins—but its short-term movement is stable enough to support the one-step-lagged control signal used by the adversary.

**Rollout-GDRO baselines and sensitivity.** For rollout allocation, the static baselines isolate a simpler alternative: use the same classifier and the same mean rollout budget, but replace the online controller with a fixed schedule. The hard-focused schedule outperforms the easy-focused inverse control (56.70 versus 54.34), supporting the view that allocation direction matters. The paper's online Rollout-GDRO result (62.27) is higher than either fixed schedule at the eval-

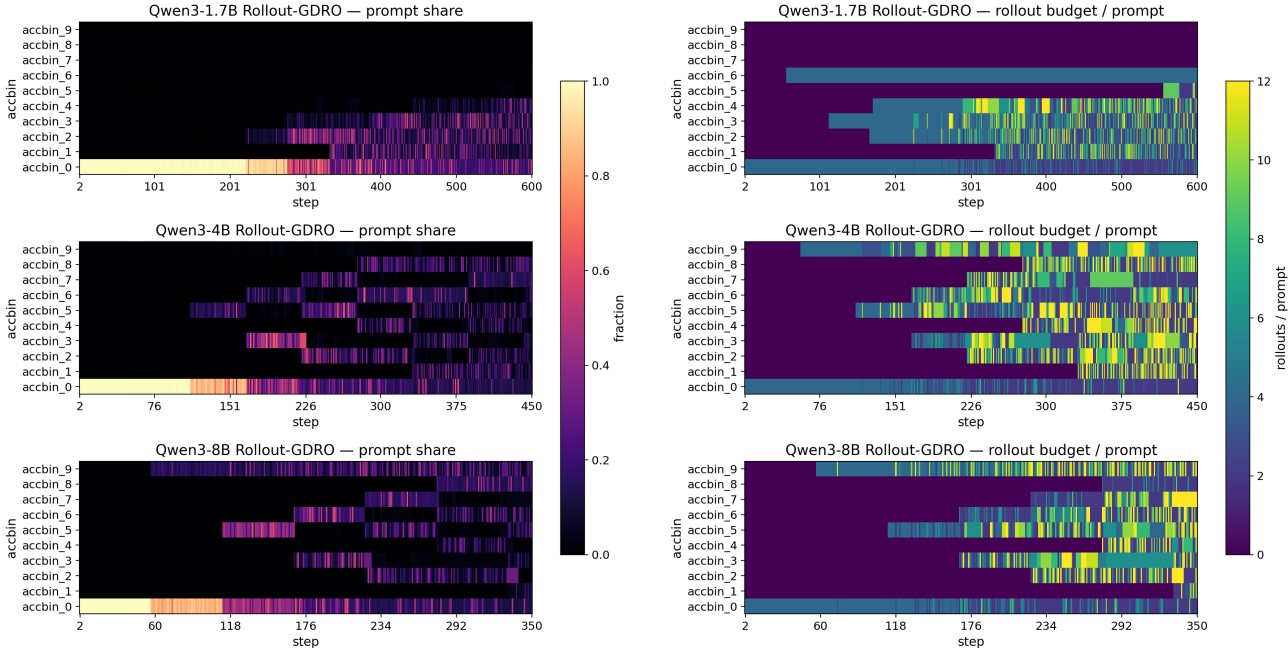

*Figure 3.* **The Budget Frontier.** A comparison of the dataset's natural difficulty distribution (Left) versus the adversarial rollout allocation (Right) for 1.7B, 4B, and 8B models. The color intensity in the Right column represents the number of rollouts assigned per prompt (from purple $\approx 2$ to yellow $\approx 12$). Note how the budgeter shifts compute intensity to the "transition zone," decoupling resource allocation from data frequency: even when hard bins are rare (dark left), they receive maximum compute (bright right).

uated checkpoints, consistent with the benefit of adapting allocation to the evolving difficulty landscape rather than committing to a hand-designed rule. The three Rollout-GDRO sensitivity variants have a narrow 0.81 pp spread, suggesting that the controller is not highly brittle to these design choices. The second-seed Rollout-GDRO run gives matching evidence at the per-benchmark level: at step 200 it improves over the GRPO baseline on four of five comparable benchmarks—MATH-500 (84.43 versus 72.05), AIME (17.46 versus 11.25), AMC (83.82 versus 60.94), and GPQA (63.69 versus 35.54), with Minerva as the exception (27.28 versus 30.48). This mirrors the directionality of the main Rollout-GDRO result and supports the conclusion that the compute-allocation gains are not a single-seed artifact.

**Runtime interpretation.** All budget claims above are mean-rollout-budget claims. They do not imply elapsed-time neutrality: the online classifier, bookkeeping, and discrete allocation logic add driver-side overhead. In our Qwen3-4B timing decomposition, median full-step times over the last 100 logged steps are 885 s for GRPO, 856 s for Prompt-GDRO, and 871 s for Rollout-GDRO, while the added advantage-stage work remains below 0.12% of total step time. The practical conclusion is therefore narrower than elapsed-time neutrality: the methods preserve the mean rollout budget, and their systems overhead should be measured in each implementation.

## 5. Conclusion

We introduced two multi-adversary GDRO frameworks for reasoning post-training that move beyond the static uniformity of standard GRPO by defining *dynamic* groups via an online difficulty classifier (stable train-time pass@$n$ binning). On this shared grouping layer, Prompt-GDRO uses an EMA-debiased GDRO-EXP3P reweighting to concentrate updates on persistently hard bins without frequency artifacts, and Rollout-GDRO uses a GDRO-EXP3P adversary with a shadow-price controller to redistribute rollouts under a fixed mean rollout budget. In this work, we evaluate Prompt-GDRO and Rollout-GDRO independently (no coupling); studying their joint dynamics is left to future work (§B). Across Qwen3-Base scales, both mechanisms preserve the mean train-time rollout budget and improve pass@8 over GRPO by up to 13.13% (Prompt-GDRO) and 10.64% (Rollout-GDRO). Targeted 4B reruns further reproduce the Prompt-GDRO gap under a second seed, show Rollout-GDRO second-seed gains on four of five comparable benchmarks, and show non-brittle sensitivity behavior.Diagnostics indicate an emergent curriculum as sampling weights and rollout budgets track the evolving reasoning frontier. These checks strengthen the empirical story: both controllers have second-seed evidence at 4B, the classifier migration is smooth enough to support stable control, and fixed hard-focused rollout schedules are stronger than easy-focused ones but remain below the adaptive Rollout-GDRO result at the evaluated checkpoints.

## Acknowledgment

We thank several anonymous ICML 2026 reviewers for their constructive comments on an earlier draft of this paper. This work was conducted while ZL, WY, and DY were at Tencent Frontier Lab (previously Tencent AI Lab) in Bellevue, WA, USA.

## Impact Statement

This paper presents work whose goal is to advance the field of Machine Learning by improving the methodological foundations of reinforcement-learning post-training for LLM reasoning through a group distributionally robust optimization (GDRO) lens. Beyond empirical gains, our theoretical contribution can be interpreted as an impact on clarity and reliability: it makes explicit which surrogate objectives the proposed controllers are implicitly optimizing, and provides rigorous (idealized) no-regret and variance-based analyses that help the community reason about when such training dynamics should behave predictably.

Improved reasoning post-training can support beneficial applications that rely on more accurate mathematical and logical outputs. As with other advances in LLM reasoning capability, it may also increase the risk of over-reliance on model outputs or misuse to generate more persuasive incorrect or harmful content; these considerations are not unique to our approach and should be addressed through careful evaluation, monitoring, and responsible deployment practices.

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

# A. Additional Related Work

Our work studies reasoning post-training at the intersection of (i) RLVR/GRPO-style policy optimization, (ii) distributionally robust optimization (DRO) and group robustness, and (iii) adaptive allocation of training-time compute.

## A.1. RLVR and Post-training for Reasoning

RL-based post-training has become central for improving reasoning behaviors in LLMs. PPO (Schulman et al., 2017) underpins RLHF-style alignment (Ouyang et al., 2022), while GRPO (Shao et al., 2024) offers a value-free, group-normalized alternative that has proven effective for verifiable-reward domains such as math. Parallel research improves reward quality and credit assignment through process supervision and step-level signals (Lightman et al., 2023; Wang et al., 2024b), as well as iterative refinement and self-improvement mechanisms (Gulcehre et al., 2023). More recently, large-scale RLVR has enabled open reasoning models trained primarily from verifiable rewards, highlighting the potential of pure RL to elicit sophisticated behaviors such as self-reflection and verification (Guo et al., 2025).

However, a growing body of work suggests that *where* RLVR learns and *how* compute is spent are both highly non-uniform. Token-level analyses indicate that RLVR gains can concentrate on a small fraction of high-entropy "forking" tokens that control reasoning branches (Wang et al., 2025), while other studies argue RLVR may implicitly incentivize correct reasoning patterns already latent in the base model (Wen et al., 2025b). At inference time, test-time scaling via longer "thinking" traces can be non-monotonic and may induce overthinking (Ghosal et al., 2025), and long-CoT reasoning models can exhibit looping pathologies under low-temperature decoding (Pipis et al., 2025). In parallel, several works explore alternative ways to trade off accuracy and compute, including budget-aware evaluation and compute-normalized comparisons (Wang et al., 2024a), and inference-time orchestration strategies that decouple accuracy from raw CoT length (Madaan et al., 2025). Our work is complementary: rather than proposing a new reward or inference strategy, we focus on *training-time* mechanisms that adaptively steer (a) which prompts are sampled and (b) how many rollouts are allocated, with the goal of improving robustness and compute efficiency.

## A.2. Distributionally Robust Optimization in Supervised Learning

DRO formalizes robustness by minimizing worst-case risk over an ambiguity set of distributions around the empirical training distribution (Ben-Tal & Nemirovski, 1999; Rahimian & Mehrotra, 2019). In supervised learning, a common choice is divergence-based ambiguity sets, yielding objectives that emphasize performance under distribution shift and provide statistical guarantees (Namkoong & Duchi, 2016; Duchi et al., 2021). Wasserstein-based DRO offers an alternative geometry with tractable reformulations and strong finite-sample guarantees (Esfahani & Kuhn, 2018). Within deep learning, GDRO (Sagawa et al., 2020) operationalizes robustness to hidden stratification and spurious correlations by optimizing the maximum loss across pre-defined groups, and has become a standard tool for improving worst-group accuracy. On the algorithmic side, group-robust learning admits a natural game-theoretic and online learning interpretation; in particular, Soma et al. (2022) connect GDRO to no-regret dynamics (e.g., EXP3 variants), which directly motivates our use of **GDRO-EXP3P** for adversarial prompt reweighting.

Our setting departs from classical supervised GDRO in two ways. First, we *do not assume static group labels*; instead we use an *online* difficulty classifier based on train-time pass@$n$ to define groups that evolve with the policy. Second, we introduce a second adversary that controls *compute allocation* (rollouts) under a budget constraint, extending the DRO perspective beyond data distribution shifts to *training-time resource shifts*.

## A.3. Robust and Distributionally Robust Reinforcement Learning

Robust RL traditionally models uncertainty in environment dynamics and seeks policies that perform well under worst-case transition perturbations, e.g., robust Markov decision processes (Iyengar, 2005; Nilim & El Ghaoui, 2005) and distributionally robust MDP formulations (Xu & Mannor, 2012). Recent work develops statistical and computational characterizations of robust RL (Panaganti et al., 2022).In contrast, our work keeps the underlying environment fixed and instead treats *prompt difficulty* and *compute allocation* as adversarially controlled quantities during LLM post-training. This yields a robustness lens that is closer to group robustness over tasks/prompts than to worst-case transition uncertainty.

### A.4. Curriculum, Adaptive Compute, and Data Value

Adaptive curricula and compute allocation are increasingly recognized as first-class components of reasoning systems. Curriculum-based post-training pipelines such as Light-R1 (Wen et al., 2025a) explicitly stage data difficulty and objectives (SFT/DPO/RL) to elicit long-CoT behaviors. At inference time, scaling laws and compute-allocation policies highlight that difficult instances require more budget, but that naive increases in "thinking" can be inefficient or unstable (Snell et al., 2024; Ghosal et al., 2025). Recent work proposes learning policies to allocate test-time compute (Setlur et al., 2025) and explores search-based or tree-structured generation to improve exploration of the reasoning space (Xie et al., 2024; Hou et al., 2025). Our **Rollout-GDRO** can be viewed as moving this idea to *training time*: we allocate rollout budgets to groups to improve gradient estimator quality under a global constraint, akin to adaptive variance reduction (Rubinstein & Kroese, 2016).

Recent work has also begun to articulate compute-optimal "RL scaling" workflows for LLM post-training by empirically studying how to allocate a fixed sampling budget across (i) the number of problems per batch, (ii) the number of parallel rollouts per problem (GRPO group size), and (iii) the number of sequential update steps. In particular, the IsoCompute Playbook reports that compute-optimal rollout parallelism can often be summarized by simple sigmoidal/logit fits as total sampling compute grows (Cheng et al., 2026). In contrast, our approach is more algorithmic: Prompt-GDRO and Rollout-GDRO adaptively reshape the effective prompt distribution and per-group compute online, without assuming a pre-fit scaling law. Concretely, we hold the mean sampling budget fixed and redistribute it across online-defined difficulty subgroups within each batch, rather than optimizing compute allocation across global axes such as problems-per-batch, rollouts-per-problem, or number of update steps. Relatedly, Qi et al. (2025) propose Budget Relative Policy Optimization (BRPO) to optimize anytime reasoning performance across varying token budgets, complementing our training-time allocation perspective.

Finally, recent theoretical discussion emphasizes that the *value of data* depends on the learner's computational constraints and even on data ordering. The notion of *epiplexity* formalizes this perspective and motivates principled data selection and dataset interventions (Finzi et al., 2026); see also community discussion (Finzi, 2026). Our work is exploratory in this broader direction: we study whether simple, online, adversarial control loops over prompt reweighting and compute allocation can serve as a practical "steering subsystem" for reasoning post-training, complementing concurrent efforts that analyze RLVR mechanisms (Wang et al., 2025; Wen et al., 2025b), rethink thinking-token tradeoffs (Madaan et al., 2025), and diagnose instabilities such as looping (Pipis et al., 2025).

## B. Limitations and Future Work

**Bridge: from two adversaries to open questions.** Our empirical results suggest that the two adversaries introduced in this work—*Prompt-GDRO* (adaptive prompt reweighting over online difficulty bins) and *Rollout-GDRO* (adaptive rollout allocation under a global compute budget)—capture complementary levers for improving reasoning post-training. Both adversaries are driven by the same online difficulty signal (stable binning via empirical train-time pass@$n$), but intervene at different points in the pipeline: Prompt-GDRO shapes the *data distribution* presented to the learner, while Rollout-GDRO shapes the *per-sample signal-to-noise ratio* of policy-gradient updates by modulating rollout counts. This coupling is central to the "beyond uniform" thesis of our framework, yet our current study is necessarily exploratory and leaves several important questions unresolved.

**Empirical scope and missing full-factorial ablations.** This paper is intended as an exploratory report to the community: we demonstrate that *distribution-shaping* tools from GDRO-style thinking can be productively instantiated inside a modern reasoning post-training stack, but we do not claim to have exhaustively optimized the design space. In particular, we have not yet performed a full factorial ablation over *all* distribution-shaping components and their interactions (e.g., prompt selection/reweighting, compute allocation, and online binning choices). Concrete future work includes:

- **Binning and classifier hyperparameters.** We used a fixed binning scheme in most experiments; broader sweeps over the number of bins, smoothing horizons, and bin-stability heuristics are needed. In preliminary sweeps we often observed performance peaking around $\approx 6$ bins, but this is not yet a robust conclusion.
- **Joint training with multiple adversaries.** Most experiments isolate Prompt-GDRO or Rollout-GDRO; a systematic study of their *joint* behavior (and staged curricula) is still missing.
- **Rollout allocator design.** We only explored a limited set of discrete rollout arms and budget schedules; future work should examine broader arm sets, alternative constrained optimizers, and adaptive rollout bounds.

**RL scaling and compute-optimal post-training.** Our experiments are conducted at a single (moderate) scale, and we do not yet understand how adversarial prompt reweighting and rollout budgeting interact with emerging "RL scaling" behavior as total post-training compute grows. Recent work (Liu et al., 2025b;a; Khatri et al., 2025; Cheng et al., 2026) study compute-optimal allocation rules and scaling behavior for RL of LLMs (e.g., by fitting simple sigmoidal/logit trends for optimal rollouts-per-problem under larger budgets). A natural next step is to evaluate whether Prompt-GDRO and Rollout-GDRO shift these compute-optimal frontiers (or their saturation points) across larger compute budgets, model sizes, and prompt mixtures.

**Adversarial game computation and systems overhead.** A practical limitation of our approach is that it introduces additional online machinery beyond standard GRPO: (i) computing and maintaining a difficulty classifier (train-time pass@$n$ statistics, stable bin assignment), (ii) updating adversarial distributions (EXP3P-style weight updates), and (iii) (for Rollout-GDRO) enforcing compute constraints while selecting discrete rollout arms. As an illustration for trade-offs, we measure the driver-side advantage stage time (`timing_s/adv`, in sec/step) and find that for the Qwen3-4B runs (mean over the last 100 logged steps after warmup) GRPO requires `0.043` sec/step, Prompt-GDRO requires `0.355` sec/step, and Rollout-GDRO requires `0.446` sec/step. This overhead is distinct from our *mean-rollout-budget* claims, which refer to the average number of generated rollouts rather than elapsed time.

We find GDRO's adversary/advantage-side bookkeeping can materially increase the driver-side advantage stage, and is one contributor to end-to-end slowdown. While each component is lightweight in isolation, their combination can create nontrivial systems overhead at scale. An important engineering direction is to design *asynchronous* and *streaming* variants (e.g., delayed bin updates, batched adversary steps, or partially offloaded bookkeeping) that preserve the core objective while minimizing training slowdown.

**Sensitivity to online binning and reward noise.** Our "group" notion is induced by an online estimator of difficulty, rather than static metadata. While this avoids reliance on brittle human-defined group labels, it also introduces potential noise sources: estimated pass@$n$ may have high variance early in training, and bin boundaries can induce discontinuous group reassignment. These effects can bias both adversaries if not handled carefully. Future work should explore principled uncertainty-aware binning (e.g., Bayesian estimators or confidence-bound assignment rules), as well as robustness to verifier calibration drift and reward-model nonstationarity. It may also be valuable to incorporate *process-level* or *stepwise* supervision signals (when available) to stabilize difficulty estimation, rather than relying solely on outcome-only pass@$n$.

**Generalization and evaluation beyond the current pipeline.** Although we report improvements on advanced benchmarks, we do not yet provide a systematic evaluation protocol for distribution shift. A natural next step is to test whether the adversarial curricula learned on one training mixture transfer to new mixtures, or whether they overfit to dataset-specific artifacts. More broadly, our method suggests a future direction where GDRO-style training is used not only to improve average in-distribution metrics, but to target robustness to hidden stratification and distribution shift (Sagawa et al., 2020; Oakden-Rayner et al., 2020). Crucially, this should be framed as future work: *out-of-distribution generalization is not a primary motivation of this paper*, but an important downstream question enabled by the methodology.

**Toward learning from the model's own experience.** A key longer-term direction is to couple our adversarial distribution shaping with *experience generation* and *continual post-training*. For example, one can imagine a closed loop where the model: (i) proposes new problems or perturbations, (ii) evaluates its own failures, and (iii) uses Prompt-GDRO/Rollout-GDRO to prioritize the resulting frontier. This connects naturally to self-training and self-improvement paradigms such as STaR-style bootstrapping (Zelikman et al., 2022), Quiet-STaR-style implicit "thinking" training (Zelikman et al., 2024), self-rewarding / judge-based optimization (Yuan et al., 2024), and RLAIF-style scalable feedback (Lee et al., 2023). Realizing such a pipeline requires addressing continual-learning issues (e.g., catastrophic forgetting (Rolnick et al., 2019)), maintaining replay buffers (Schaul et al., 2015), and preventing reward hacking under self-generated supervision.

**Beyond exponential-weights GDRO: richer ambiguity sets and scalable solvers.** Our current instantiation uses exponential-weights style updates over a discrete set of groups/arms. However, the broader DRO literature offers many alternative ambiguity sets and solution methods that may be better suited for future scaling: $f$-divergence DRO admits stochastic-gradient formulations (Namkoong & Duchi, 2016), and recent work studies computationally efficient large-scale solvers for DRO objectives such as CVaR and $\chi^2$-based uncertainty sets (Levy et al., 2020). Wasserstein DRO provides a complementary metric-based robustness lens with tractable reformulations and finite-sample guarantees (Esfahani & Kuhn,

2018). On the RL side, robust MDP formulations (Iyengar, 2005; Nilim & El Ghaoui, 2005) and scalable $\phi$-divergence regularization approaches (Panaganti et al., 2024) suggest additional ways to model uncertainty and allocate resources under environment shift. A concrete research agenda is to identify which ambiguity sets best correspond to the *operational* failure modes of reasoning post-training (e.g., verifier mismatch, data mixture drift, or hard-sample scarcity), and to develop scalable solvers compatible with modern LLM training.

**Beyond robustness: adversarial reasoning objectives for safety and risk.** Finally, our adversaries currently act on *what* is trained (prompt distribution) and *how intensively* it is trained (rollout budgets), but not on richer forms of adversarial reasoning objectives. An important future direction is to design adversaries that target *specific* reasoning desiderata beyond accuracy, such as safety, risk avoidance, and constraint satisfaction. This connects to alignment frameworks that rely on rule-based or AI-generated feedback (Bai et al., 2022; Lee et al., 2023), as well as risk-sensitive optimization objectives. We view this as a distinct problem from classical DRO: rather than protecting against distributional uncertainty alone, the goal becomes to adversarially surface and correct *undesirable reasoning behaviors* (e.g., unsafe tool use, brittle shortcuts, or overconfident hallucinations) under realistic deployment constraints.

# C. Experiment Details

In this section, we detail the experimental setup, including the shared optimization hyperparameters and the specific configurations for our adversarial mechanisms. All experiments were conducted using the Qwen3-Base model family (1.7B, 4B, 8B) using BFloat16 (BF16) mixed precision and FlashAttention 2. In addition to the details below, we provide the code and configuration files used to reproduce these results in the supplementary materials.

**Evaluation benchmarks.** In the main results table, "MATH-500" refers to the HuggingFaceH4/MATH-500 benchmark.[2]

## C.1. Shared Training Hyperparameters

All methods (GRPO Baseline, Prompt-GDRO, and Rollout-GDRO) utilize a common post-training foundation based on the Group Relative Policy Optimization (GRPO) objective.

### C.1.1. OPTIMIZATION & ARCHITECTURE

- **Global Train Batch Size**: 256

- **Global Validation Batch Size**: 128

- **Total Training Steps**: 1000

- **Random Seed**: The main scale sweep uses the paper's primary seed; Section 4.4 adds Qwen3-4B second-seed checks for both Prompt-GDRO and Rollout-GDRO, together with targeted sensitivity runs.

- **Optimizer**: AdamW

- **Actor Learning Rate**: $1 \times 10^{-6}$

- **KL Penalty Coefficient** ($\beta_{\text{KL}}$): 0.001

- **PPO Clip Range**: $[1 - \epsilon_{\text{low}}, 1 + \epsilon_{\text{high}}]$ where $\epsilon_{\text{low}} = 0.2$, $\epsilon_{\text{high}} = 0.28$

- **Advantage Normalization**: Yes (Normalized by group standard deviation)

- **Advantage Clipping**: $[-5, 5]$

### C.1.2. ROLLOUT GENERATION

- **Inference Engine**: vLLM

- **Training Rollouts per Prompt** ($G$): 4 (Base setting)

---

[2] https://huggingface.co/datasets/HuggingFaceH4/MATH-500

- **Validation Rollouts per Prompt**: 8

- **Sampling Temperature**: 0.6 (Training)

- **Top-p**: 0.8

- **Top-k**: 20

- **Reward**: Verifiable math correctness with $r(x, y) \in \{-1, +1\} \subset [-1, 1]$, implemented via the `math/math-dapo` modules in `verl`.

## C.2. Adversarial Configuration

Our Multi-Adversary framework introduces specific hyperparameters for the **EXP3P** algorithms governing data sampling and compute allocation.

### C.2.1. PROMPT-GDRO (THE DATA ADVERSARY)

This mechanism reweights the prompt distribution based on the intensive difficulty of online groups.

- **Grouping Mechanism**: Online train-time Pass@$n$ (10 bins; $n = 4$ fixed; sliding window $H \in [50, 100]$; plug-in estimator from Section 3)

- **Adversary Algorithm**: EMA-Debiased GDRO-EXP3P

- **Adversary Learning Rate** ($\eta_q$): 0.65

- **Exploration Rate** ($\gamma$): 0.01

- **Score EMA Decay** ($\beta$): 0.12

- **Max Class Weight Cap**: 15.0

- **Loss Normalization**: Normalized by class share (to prevent frequency bias)

### C.2.2. ROLLOUT-GDRO (THE COMPUTE ADVERSARY)

This mechanism allocates discrete rollout counts $n_b$ to minimize gradient variance under a global budget constraint.

- **Grouping Mechanism**: Online train-time Pass@$n$ (10 bins, edges at $0.1, 0.2, \ldots, 0.9$; computed from the rollouts actually generated, so $n$ is bin-dependent under Rollout-GDRO; sliding window $H \in [50, 100]$; plug-in estimator from Section 3)

- **Rollout Arm Range**: $n \in [n_{\min}, n_{\max}]$ where $n_{\min} = 2, n_{\max} = 12$ (Multiplier $3.0\times$ base)

- **Global Budget Constraint** ($\bar{n}$): 4 rollouts (average per prompt)

- **Dual Learning Rate** ($\alpha_\mu$): 0.05

- **Arm Learning Rate** ($\eta$): 0.65

- **Arm Exploration Rate** ($\gamma$): 0.01

- **Arm Score EMA Decay**: 0.4

- **Budget Matching**: Exact constrained selection via Dynamic Programming

**Systems overhead.** The online train-time pass@$n$ grouping/bookkeeping and (for Rollout-GDRO) exact dynamic-programming budget matching introduce additional driver-side overhead. These overheads are separate from mean-rollout-budget preservation and should be interpreted as implementation-dependent systems costs; Section 4.4 reports the corresponding Qwen3-4B timing decomposition.

# D. Additional Qualitative Analysis

This section collects additional qualitative figures and supporting discussion deferred from Section 4 to satisfy the main-paper page limit.

## D.1. Prompt-GDRO: The Dynamics of Difficulty

**Capacity-Dependent Distribution Shift.** The training dynamics reveal a stark correlation between model capacity and the "velocity" of learning, as quantified by the macro-level metrics in Figure 4.

- **Qwen3-1.7B (Top Row, Fig 2):** The model exhibits high inertia. While the dataset mass often lags in `accbin_0`, the scalar summary (Figure 4, top panel) shows the mean bin index steadily climbing past 2.0. The **mass in bins** $\geq 3$ trace reveals that even this smaller model is successfully pushed out of the trivial zone, preventing the stagnation typical of uniform baselines.
- **Qwen3-4B (Middle Row, Fig 2):** This scale occupies a "Goldilocks zone." The data distribution shows a steady migration to intermediate bins. The adversarial weights form a distinct, high-intensity **diagonal frontier** that leads the data distribution. By Step 366, the weight entropy stabilizes, indicating the adversary has locked into a high-precision curriculum targeting specific intermediate failure modes.
- **Qwen3-8B (Bottom Row, Fig 2):** The largest model exhibits a rapid collapse of the unsolved mass. The scalar summaries show the **mass in bins** $\geq 8$ (dashed lines) spiking early, confirming that for capable models, the adversary aggressively focuses on the "last mile" of robustness—solving the few remaining hard cases—rather than wasting capacity on solved arithmetic.

**Visualizing the Wave of Progress.** Figure 5 offers discrete checkpoints that clarify the exact distributional shape at key training intervals.

- **The Heavy Tail (1.7B):** Even at the final step (Step 605), the 1.7B model retains a significant plurality of mass ($\approx 45\%$) in `accbin_0`. The distribution remains right-skewed, indicating the "reasoning frontier" is anchored in fundamental difficulties.
- **The High-Entropy Plateau (4B):** By Step 366, the 4B model allocates over 50% of its probability mass to the intermediate `accbin_5–accbin_7` range. This "plateau" represents a diverse curriculum where the model simultaneously refines intermediate concepts and attempts advanced problems.
- **The Traveling Peak (8B):** The 8B model demonstrates a clear "wave" dynamic. The peak of the distribution physically travels from left to right. By Step 430, the mass in `accbin_0` has virtually vanished, and the bulk of the sampling budget is dedicated to `accbin_8` and `accbin_9`. This confirms that for sufficient capacity, the primary challenge shifts rapidly from *correctness* to *robustness*, necessitating the dynamic budget reallocation our method provides.

**Adversary lead–lag.** The heatmaps in Figure 2 suggest that the adversarial weights form a "frontier" that *precedes* the empirical data distribution. We quantify this intuition with a lightweight lead–lag proxy computed from logged bin statistics. Let $q_t \in \Delta_B$ denote the empirical *prompt share* over bins at training step $t$, and let $\hat{w}_t \in \Delta_B$ denote the *normalized* Prompt-GDRO weights across bins at the same step (the weight-only distribution, not reweighted by $q_t$). Define the mean bin index under the data and under the weights as

$$\mu_{\text{data}}(t) \triangleq \sum_{b=1}^{B} b\, q_t(b), \qquad \mu_{\text{weight}}(t) \triangleq \sum_{b=1}^{B} b\, \hat{w}_t(b), \tag{22}$$

and the lead–lag gap $\Delta\mu(t) \triangleq \mu_{\text{weight}}(t) - \mu_{\text{data}}(t)$. A positive $\Delta\mu(t)$ indicates that the adversary's weight distribution is shifted toward higher-index bins than the empirical prompt share. Under our convention that `accbin_0` corresponds to the *lowest* train-time pass@$n$ interval and larger indices correspond to *higher* train-time pass@$n$ (more solvable under the train-time rollout budget) prompts, this means the adversary emphasizes the current *learnable frontier* rather than simply mirroring the bulk of unsolved mass. Figure 6 (left) shows that $\Delta\mu(t)$ is strongly positive early in training and decays over time, with faster decay at larger model scales. For Qwen3-8B, $\Delta\mu(t)$ eventually becomes slightly negative, consistent with the data distribution migrating quickly into high pass@$n$ bins while the adversary maintains pressure on the remaining low-pass@$n$ cases. Overall, the decay of $\Delta\mu(t)$ provides a compact scalar signature of the "traveling wave" curriculum: the adversary leads and the data distribution catches up as the policy improves.

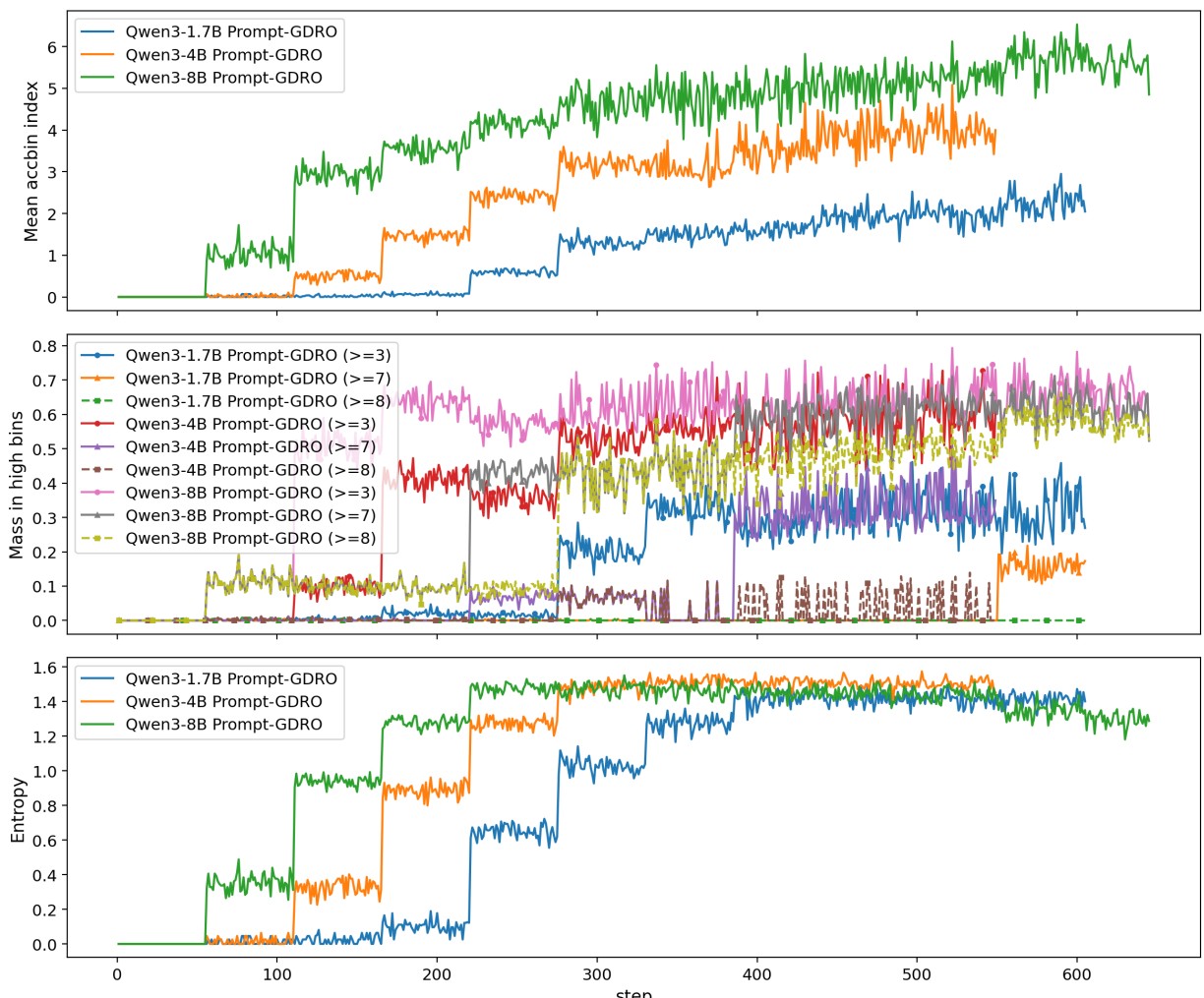

*Figure 4.* **Training Dynamics Quantified (Prompt-GDRO).** (Top) The Mean Accuracy Bin Index tracks the rising difficulty of targeted prompts. (Middle) The fraction of prompts in difficulty bands $\geq 3$ (solid) and $\geq 8$ (dashed) highlights scaling laws: 1.7B struggles to clear the $\geq 3$ bar, while 8B rapidly saturates even the $\geq 8$ band. (Bottom) Entropy metrics confirm that the adversary maintains a diverse portfolio of difficulty, preventing mode collapse to a single bin.

## D.2. Rollout-GDRO: Adaptive Compute Allocation

### D.2.1. MACRO-DYNAMICS OF ALLOCATION

To quantify these trends, Figure 7 presents the macro-level training dynamics. The four stacked traces—Mean Accuracy Bin Index, High-Bin Mass, High-Bin Budget Share, and Entropy—provide a unified view of how Rollout-GDRO migrates compute toward harder domains.

The "Mass in High Bins" trace is particularly revealing. It serves as direct evidence of the DRO objective in action: as the models improve, the adversary steadily reallocates the fixed global budget toward bins $\geq 7$. This reallocation correlates directly with the inflection points observed in the *evaluation* pass@8 accuracy tables. Furthermore, the shared axes highlight distinct scaling trends: the 8B model (green line) learns to push its budget into high-difficulty bins significantly earlier than the 1.7B model (blue line), validating that larger capacity enables more aggressive curriculum acceleration.

**Variance-aware compute efficiency.** Beyond shifting budget toward harder bins, we can directly test whether the rollout adversary reduces the *uncertainty* of the bin-wise training signal at fixed compute. Let $n_b(t)$ denote the realized number of rollouts per prompt allocated to bin $b$ at step $t$, and let $q_t(b)$ denote the prompt share. Using an offline bin-wise variability proxy $\hat{\sigma}_b$ (estimated once from the training logs as the empirical standard deviation of a per-prompt GRPO signal within

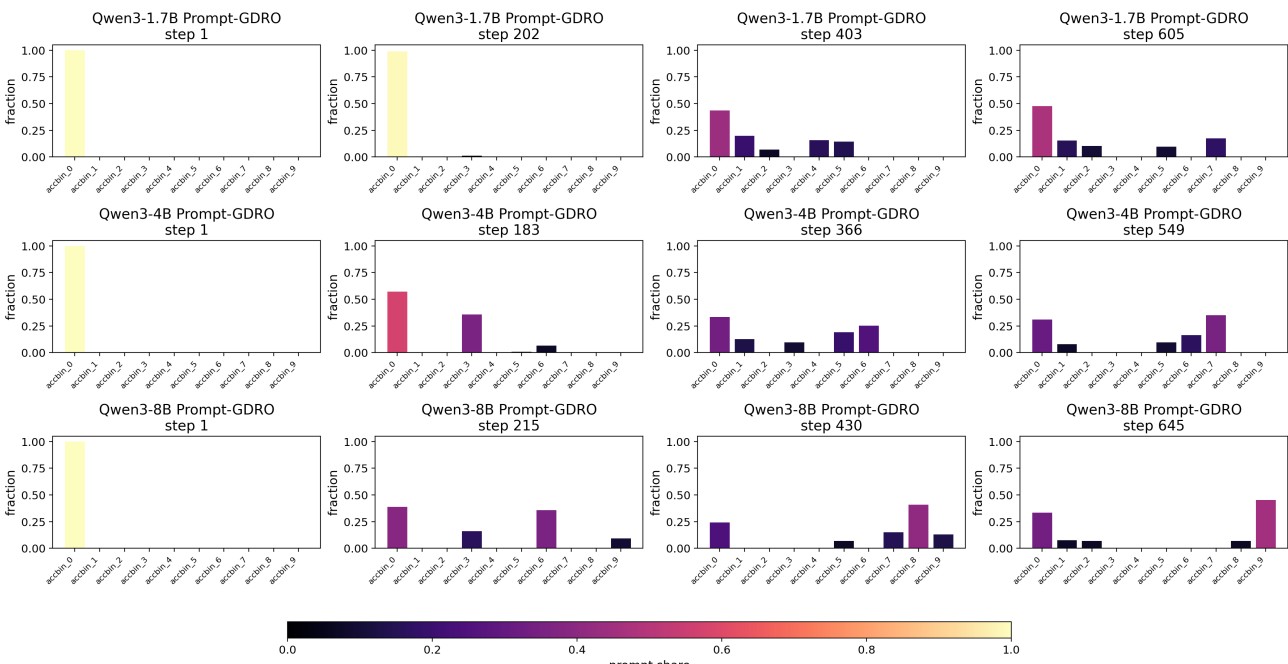

*Figure 5.* **Snapshots of the Learning Distribution.** The probability mass of the training set across difficulty bins at four distinct training stages (Start, Early-Mid, Late-Mid, End). These publication-friendly checkpoints reveal the exact shape of the curriculum: note how the 4B model (Middle Row) transitions from a uniform start to a heavy emphasis on `accbin_5–accbin_7` by mid-training, whereas the 8B model (Bottom Row) shifts almost its entire mass to the hardest bins by the final step.

each bin, e.g., the prompt-level loss $\ell(x;\theta)$). Concretely, we log the prompt-level signal for each prompt throughout training together with its current bin assignment $g_t(x)$, pool these logged values by bin over the full run, and compute $\hat{\sigma}_b$ as the sample standard deviation of the pooled set. This yields a fixed (static) $\hat{\sigma}_b$ per run; it is used only as a post-hoc diagnostic and is not consumed by the Rollout-GDRO controller. we define a weighted standard-error proxy

$$\text{WSE}(t) \triangleq \sum_{b=1}^{B} q_t(b) \frac{\hat{\sigma}_b}{\sqrt{n_b(t)}}. \tag{23}$$

We compare against a compute-matched uniform-rollout baseline by setting $n_b(t) \equiv \bar{n}$ for all bins. This uniform allocation satisfies the same mean-rollout constraint $\sum_b q_t(b)\, n_b(t) = \bar{n}$ at each step and thus matches overall sampling compute. As shown in Figure 6 (right), Rollout-GDRO consistently attains lower $\text{WSE}(t)$ than the uniform baseline throughout training. Averaged over the plotted horizon, this corresponds to relative reductions of **37.1%** (1.7B), **22.6%** (4B), and **33.4%** (8B), supporting the interpretation that Rollout-GDRO improves gradient information efficiency by allocating more rollouts to high-variance bins.

### D.2.2. DISCRETE ECONOMIC PHASES

Finally, Figure 8 decomposes the continuous training process into discrete "chapters" of the curriculum. These snapshots clarify the magnitude of the adversarial intervention at key training stages (Start, Early, Mid, Late).

The paired bars (Dataset Share vs. Rollout Budget) illustrate a massive **Multiplier Effect**. For example, at Step 300, the 4B model allocates over $80\%$ of its compute budget to bins $\geq 5$, despite these bins constituting less than $20\%$ of the training data. This confirms that our method creates a highly non-uniform economic policy that gives $5$–$10\times$ more rollouts to the "reasoning frontier" than a uniform baseline would. This behavior is model-specific: the 8B row shows an even faster shift, embracing high-bin budgeting early in training (Step 118), which aligns with the rapid saturation of easy tasks observed in our qualitative analysis.

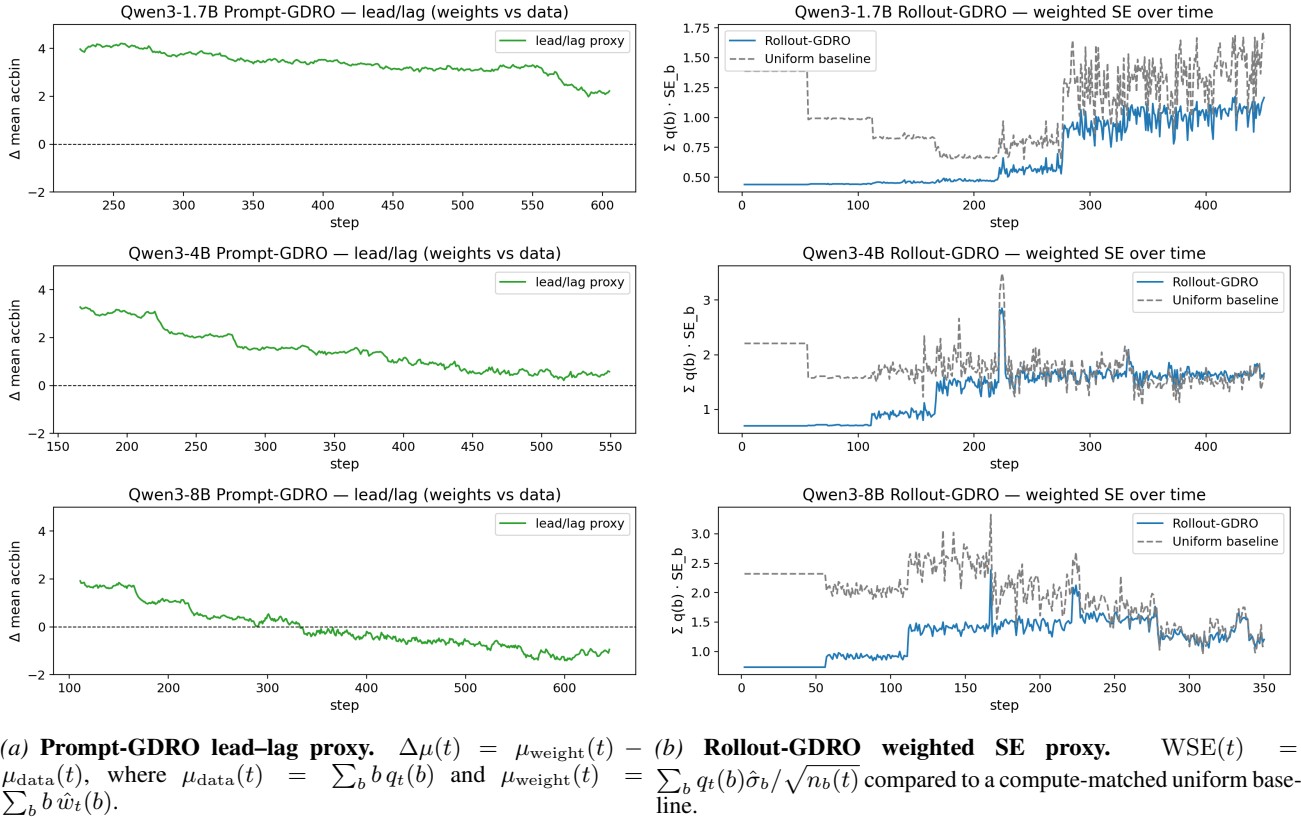

*(a)* **Prompt-GDRO lead–lag proxy.** $\Delta\mu(t) = \mu_{\text{weight}}(t) - \mu_{\text{data}}(t)$, where $\mu_{\text{data}}(t) = \sum_b b\, q_t(b)$ and $\mu_{\text{weight}}(t) = \sum_b b\, \hat{w}_t(b)$.

*(b)* **Rollout-GDRO weighted SE proxy.** $\text{WSE}(t) = \sum_b q_t(b)\hat{\sigma}_b/\sqrt{n_b(t)}$ compared to a compute-matched uniform baseline.

*Figure 6.* **Two diagnostics for the two adversaries.** (Left) Prompt-GDRO weights initially lead the empirical data distribution in bin index (a "learnable frontier") and progressively align as the prompt-share distribution catches up. (Right) Rollout-GDRO reduces an offline weighted standard-error proxy relative to a uniform allocation with the same mean rollout budget, consistent with variance-aware compute allocation.

# E. Main Theoretical Results: A Game-and-Variance View

This section develops a unified theoretical lens for the *two adversarial controllers* in our framework: (i) **Prompt-GDRO**, which adaptively reshapes the prompt distribution to emphasize difficult bins, and (ii) **Rollout-GDRO**, which adaptively reallocates rollouts across bins to reduce estimation noise under a compute budget. Our goal is *not* to provide deep-network convergence guarantees for GRPO, but rather to (a) formalize the surrogate objectives implicitly optimized by these controllers, and (b) connect their update rules to standard no-regret / mirror-descent analyses that explain the qualitative behaviors observed empirically (e.g., "traveling waves" and staircase compute allocation).

A condensed statement of these results (omitting most proofs) appears in Section 3; this appendix provides complete statements and proofs for reference.

Throughout, we adopt the GDRO formulation from Section 2 (Eq. (9)). Let $g(x) \in \{1, \ldots, B\}$ be the (online) grouping rule from Section 3, and define the group losses below. We use $\{L_b\}_{b=1}^B$ primarily in the Prompt-GDRO analysis; Rollout-GDRO instead optimizes a separate budgeted variance objective.

$$L_b(\theta) \triangleq \mathbb{E}[\ell(x; \theta) \mid g(x) = b], \qquad b \in \{1, \ldots, B\}, \tag{24}$$

where $\ell(x; \theta)$ is the prompt-level GRPO loss from Section 2.2.[3]

---

[3]In reward maximization form, one may take $L_b(\theta) = -J_b(\theta)$, where $J_b$ is the group-conditional expected reward (this sign convention is standard in RL; see, e.g., Agarwal et al., 2019).

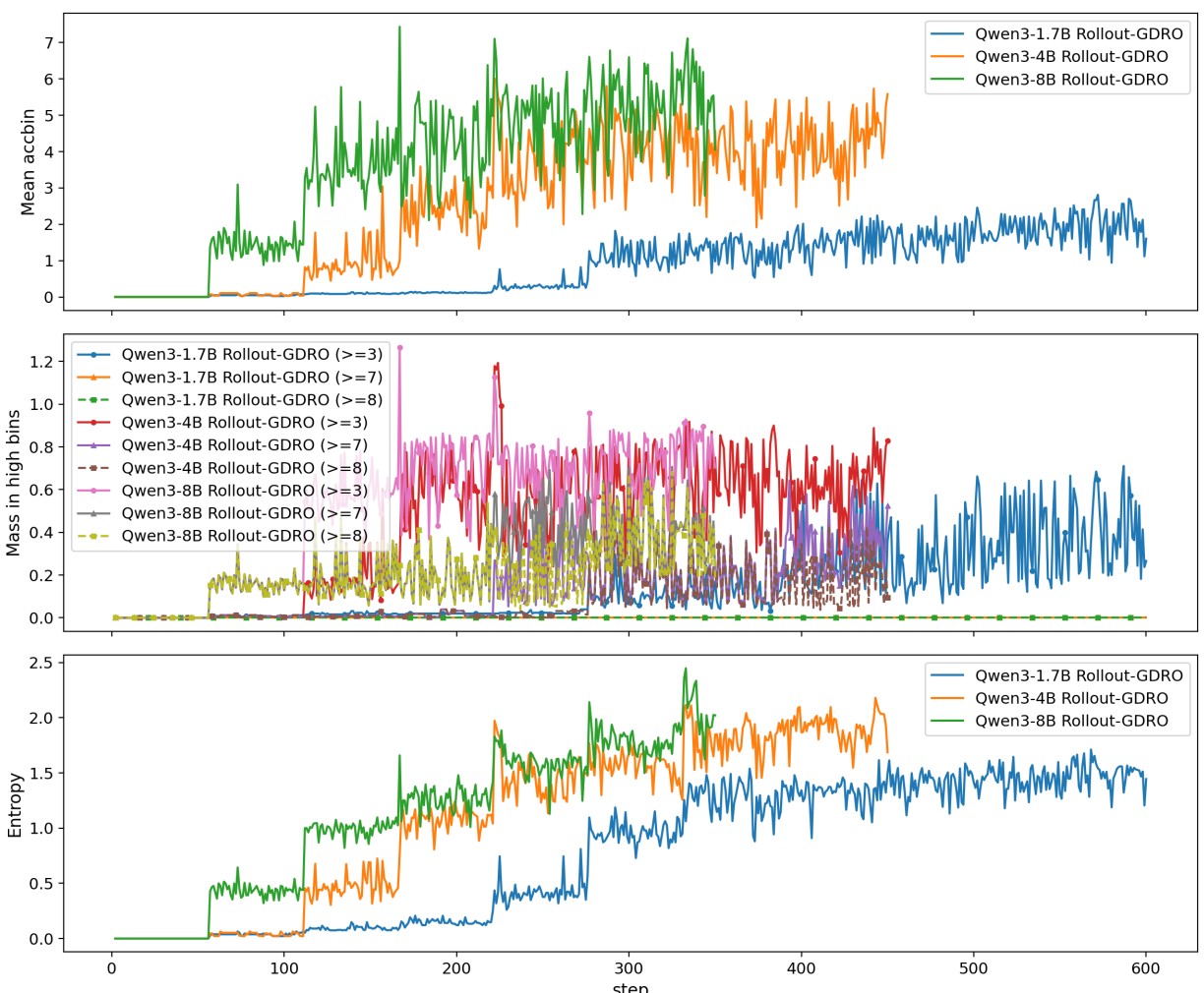

*Figure 7.* **Macro-Level Allocation Dynamics (Rollout-GDRO).** (Top) The Mean Accuracy Bin Index tracks the rising difficulty of targeted prompts. (Middle) The "Mass in High Bins" trace serves as quantitative evidence of the rollout adversary: it shows the dual variable mechanism keeping the total budget fixed while aggressively reweighting toward hard groups ($\geq$ accbin_8). (Bottom) Entropy metrics confirm that the 8B model (green) sustains a diverse allocation strategy even as it conquers lower difficulties, contrasting with the slower migration of the 1.7B model (blue).

### E.1. Prompt-GDRO as Entropic GDRO and No-Regret Game Dynamics

We start from the canonical finite-group robust objective

$$\min_{\theta}\ \max_{q\in\Delta_B}\ f(\theta,q), \qquad f(\theta,q) \triangleq \sum_{b=1}^{B} q(b)\, L_b(\theta), \tag{25}$$

where $\Delta_B$ is the probability simplex over $B$ groups. The inner maximization selects a worst-case mixture of groups, while the outer minimization trains a policy robust to this mixture.

E.1.1. ENTROPY-REGULARIZED INNER MAXIMIZATION AND THE LOG-SUM-EXP SURROGATE

A recurring theme in this paper is that our adversaries are implemented by *exponential-weights* updates (i.e., entropic mirror descent/ascent). A key consequence is that the exact max/min over the simplex is replaced by an *entropy-regularized* version, yielding a smooth "soft" worst-group objective.

The next lemma is a standard variational identity (often called the Gibbs variational principle / the convex conjugacy between

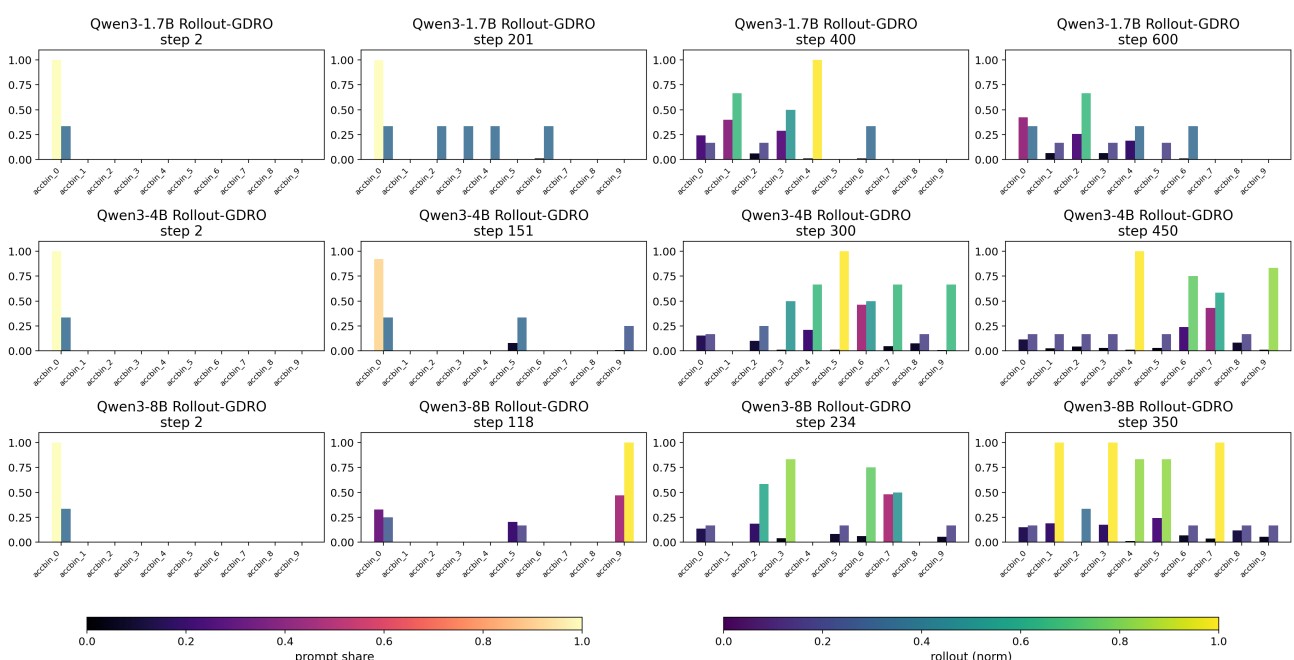

*Figure 8.* **Snapshots of the Allocation Economy.** Paired bars at four canonical steps showing the dataset share (dark blue) versus the normalized rollout budget (light blue) for each bin. This visualizes the **"Multiplier Effect"**: by Step 300, the 4B model allocates $> 80\%$ of its budget to `accbin_5+`, even though these bins contain $< 20\%$ of the data. This explicitly demonstrates how Rollout-GDRO amplifies the signal from rare, high-value prompts.

log-sum-exp and negative entropy); see, e.g., Boyd & Vandenberghe (2004) or Wainwright & Jordan (2008). We include a proof for completeness.

**Lemma E.1** (Entropy-regularized maximum and log-sum-exp). *For any $v \in \mathbb{R}^m$ and $\eta > 0$,*

$$\max_{p \in \Delta_m} \left\{ \langle p, v \rangle + \frac{1}{\eta} H(p) \right\} = \frac{1}{\eta} \log \left( \sum_{i=1}^{m} e^{\eta v_i} \right), \tag{26}$$

*where $H(p) \triangleq -\sum_{i=1}^{m} p_i \log p_i$ is the Shannon entropy. Moreover, the maximizer is unique and equals the softmax distribution*

$$p_i^{\star}(v) = \frac{e^{\eta v_i}}{\sum_{j=1}^{m} e^{\eta v_j}} \qquad (i = 1, \ldots, m). \tag{27}$$

*Proof.* Consider the Lagrangian for maximizing $\sum_i p_i v_i - \frac{1}{\eta} \sum_i p_i \log p_i$ subject to $\sum_i p_i = 1$ and $p_i \geq 0$. At an interior optimum (which holds here because the entropy term makes the objective *strictly concave* and favors $p_i > 0$), stationarity gives

$$v_i - \frac{1}{\eta}(\log p_i + 1) = \lambda \qquad \text{for all } i,$$

Here $\lambda \in \mathbb{R}$ is the Lagrange multiplier for the equality constraint $\sum_i p_i = 1$ (so its sign is unconstrained); rearranging yields $p_i \propto e^{\eta v_i}$. Normalizing yields (27). Plugging (27) into the objective gives

$$\sum_i p_i^{\star} v_i + \frac{1}{\eta} H(p^{\star}) = \frac{1}{\eta} \log \left( \sum_{i=1}^{m} e^{\eta v_i} \right),$$

which proves (26). Uniqueness follows from strict concavity of $H(\cdot)$ on $\Delta_m$. $\qquad \square$

**Corollary E.2** (Smooth approximation quality). *For any $v \in \mathbb{R}^m$ and $\eta > 0$,*

$$\max_i v_i \leq \frac{1}{\eta} \log \left( \sum_{i=1}^{m} e^{\eta v_i} \right) \leq \max_i v_i + \frac{\log m}{\eta}. \tag{28}$$

*Proof.* Let $v_{\max} = \max_i v_i$. Then $\sum_i e^{\eta v_i} \geq e^{\eta v_{\max}}$, giving the lower bound. Also $\sum_i e^{\eta v_i} \leq m e^{\eta v_{\max}}$, giving the upper bound. $\qquad \square$

*Remark* E.3 (KL-robust view and "implicit" regularization). Let $u$ be the uniform distribution over $[m]$. Using $\mathrm{KL}(p\|u) = \sum_i p_i \log p_i + \log m$ and $H(p) = -\sum_i p_i \log p_i$, (26) is equivalently

$$\frac{1}{\eta} \log \left( \sum_{i=1}^{m} e^{\eta v_i} \right) = \frac{\log m}{\eta} + \max_{p \in \Delta_m} \left\{ \langle p, v \rangle - \frac{1}{\eta} \mathrm{KL}(p\|u) \right\}. \tag{29}$$

Thus, the softmax distribution in (27) can be interpreted as the optimizer of a *KL-penalized* DRO objective. In our implementation, this entropy/KL term is not added explicitly to (25); it appears implicitly because the adversary is implemented by entropic mirror ascent (exponential weights). Corollary E.2 quantifies the resulting max-gap: the soft objective approximates the hard max within $\log m / \eta$.

**Entropic GDRO surrogate.** Applying Lemma E.1 with $m = B$ and $v = L(\theta) \triangleq (L_1(\theta), \dots, L_B(\theta))$ shows that the entropy-regularized inner problem in (25) yields the smooth robust surrogate

$$\mathcal{R}_\eta(\theta) \triangleq \max_{q \in \Delta_B} \left\{ \sum_{b=1}^{B} q(b) L_b(\theta) + \frac{1}{\eta} H(q) \right\} = \frac{1}{\eta} \log \left( \sum_{b=1}^{B} e^{\eta L_b(\theta)} \right). \tag{30}$$

**Corollary E.4** (Entropic GDRO as a surrogate for worst-group loss). *For any $\theta$, the entropic surrogate in (30) satisfies*

$$\max_{b \in [B]} L_b(\theta) \leq \mathcal{R}_\eta(\theta) \leq \max_{b \in [B]} L_b(\theta) + \frac{\log B}{\eta}. \tag{31}$$

*Equivalently, $\mathcal{R}_\eta(\theta)$ is the value of the entropy-regularized variant of the inner maximization in (25).*

*Proof.* Apply Corollary E.2 to $v = L(\theta)$ with $m = B$ and note that $\max_i v_i = \max_{b \in [B]} L_b(\theta)$. $\qquad \square$

By Corollary E.4, $\mathcal{R}_\eta(\theta)$ is a differentiable approximation to $\max_b L_b(\theta)$.

**Lemma E.5** (Gradient of the entropic worst-group surrogate). *Assume each $L_b(\theta)$ is differentiable. Define $q_\eta(\cdot; \theta) \in \Delta_B$ by*

$$q_\eta(b; \theta) \triangleq \frac{\exp(\eta L_b(\theta))}{\sum_{j=1}^{B} \exp(\eta L_j(\theta))}. \tag{32}$$

*Then $\nabla_\theta \mathcal{R}_\eta(\theta) = \sum_{b=1}^{B} q_\eta(b; \theta) \nabla_\theta L_b(\theta)$.*

*Proof.* This identity follows directly from *Danskin's theorem* applied to the inner maximization in (30). Indeed, $\Delta_B$ is compact and the entropy regularizer makes the inner problem strictly concave in $q$, hence the maximizer $q_\eta(\cdot; \theta)$ is unique. We include the short closed-form differentiation of (30) below for completeness:

$$\nabla_\theta \mathcal{R}_\eta(\theta) = \frac{1}{\eta} \cdot \frac{\sum_{b=1}^{B} e^{\eta L_b(\theta)} \eta \nabla_\theta L_b(\theta)}{\sum_{j=1}^{B} e^{\eta L_j(\theta)}} = \sum_{b=1}^{B} q_\eta(b; \theta) \nabla_\theta L_b(\theta).$$

$\qquad \square$

**Interpretation.** The distribution $q_\eta(\cdot; \theta)$ in (32) is exactly the entropy-regularized best response of the adversary to $\theta$: it solves $\arg\max_{q \in \Delta_B} \{ \sum_b q(b) L_b(\theta) + \frac{1}{\eta} H(q) \}$. Thus $q_\eta(\cdot; \theta)$ is a smooth proxy for the hard worst-group selector in (25): as $\eta \to \infty$, $q_\eta(\cdot; \theta)$ concentrates on (ties among) $\arg\max_b L_b(\theta)$, whereas for finite $\eta$ it spreads mass across near-worst groups. This is the same "soft" best response tracked online by the exponential-weights update used in Prompt-GDRO.

E.1.2. NO-REGRET DYNAMICS IMPLY APPROXIMATE ROBUST OPTIMALITY

We now connect Prompt-GDRO to standard min–max optimization via no-regret dynamics. The main message is classical: if the learner (policy) and adversary (group distribution) each run a no-regret algorithm, then their time averages approach an approximate saddle point of (25). We provide a self-contained theorem in an idealized convex bounded regime, mainly to make the core maximin/no-regret logic behind Prompt-GDRO explicit.

**Assumption E.6** (Convex bounded regime). *Assume $\Theta \subset \mathbb{R}^d$ is convex and compact with diameter $D$ in $\ell_2$. Assume each group loss $L_b(\theta)$ is convex in $\theta$, differentiable, and $G$-Lipschitz: $\|\nabla_\theta L_b(\theta)\|_2 \leq G$ for all $\theta \in \Theta$. Assume the losses are bounded: $0 \leq L_b(\theta) \leq M$ for all $b, \theta$.*

**Theorem E.7** (No-regret dynamics yield an approximate GDRO solution). *Consider the zero-sum game (25) with payoff $f(\theta, q) \triangleq \sum_{b=1}^{B} q(b) L_b(\theta)$. Let Assumption E.6 hold. Suppose we run $T$ rounds of:*

1. *(**Learner**) Online gradient descent on $\theta$ with step size $\eta_\theta$: $\theta_{t+1} = \Pi_\Theta(\theta_t - \eta_\theta g_t)$, where $g_t$ is a (possibly stochastic) subgradient satisfying $\mathbb{E}[g_t \mid \theta_t, q_t] = \nabla_\theta f(\theta_t, q_t)$ and a conditional second-moment bound $\mathbb{E}[\|g_t\|_2^2 \mid \theta_t, q_t] \leq G_{\text{sg}}^2$.*
2. *(**Adversary**) Exponentiated-gradient mirror ascent on $q$ with step size $\eta_q$: $q_{t+1}(b) \propto q_t(b) \exp(\eta_q \hat{L}_{t,b})$, where $\hat{L}_{t,b}$ satisfies $\mathbb{E}[\hat{L}_{t,b} \mid \theta_t] = L_b(\theta_t)$ and $0 \leq \hat{L}_{t,b} \leq M$ a.s.*

*Let $\bar{\theta} \triangleq \frac{1}{T} \sum_{t=1}^{T} \theta_t$ and $\bar{q} \triangleq \frac{1}{T} \sum_{t=1}^{T} q_t$. Then the expected saddle-point gap satisfies*

$$\mathbb{E}\Big[ \max_{q \in \Delta_B} f(\bar{\theta}, q) - \min_{\theta \in \Theta} f(\theta, \bar{q}) \Big] \leq \frac{D^2}{2\eta_\theta T} + \frac{\eta_\theta G_{\text{sg}}^2}{2} + \frac{\log B}{\eta_q T} + \frac{\eta_q M^2}{8}. \tag{33}$$

*Consequently, since $\max_{q \in \Delta_B} f(\bar{\theta}, q) = \max_b L_b(\bar{\theta})$ and $\min_\theta \max_q f(\theta, q)$ is the GDRO optimum,*

$$\mathbb{E}\big[ \max_b L_b(\bar{\theta}) \big] \leq \min_{\theta \in \Theta} \max_b L_b(\theta) + \frac{D^2}{2\eta_\theta T} + \frac{\eta_\theta G_{\text{sg}}^2}{2} + \frac{\log B}{\eta_q T} + \frac{\eta_q M^2}{8}. \tag{34}$$

*Proof.* We prove (33) by combining standard regret bounds for OGD (learner) and exponential-weights on the simplex (adversary); see, e.g., Bubeck, 2015; Hazan, 2016; Cesa-Bianchi & Lugosi, 2006. We follow these textbook proofs and include the derivation here mainly to track constants.

**Step 1: learner regret.** By non-expansiveness of Euclidean projection and the standard OGD one-step inequality, for any $\theta \in \Theta$,

$$\|\theta_{t+1} - \theta\|_2^2 \leq \|\theta_t - \eta_\theta g_t - \theta\|_2^2 = \|\theta_t - \theta\|_2^2 - 2\eta_\theta \langle g_t, \theta_t - \theta \rangle + \eta_\theta^2 \|g_t\|_2^2.$$

Rearranging and summing over $t = 1, \ldots, T$ yields

$$\sum_{t=1}^{T} \langle g_t, \theta_t - \theta \rangle \leq \frac{\|\theta_1 - \theta\|_2^2}{2\eta_\theta} + \frac{\eta_\theta}{2} \sum_{t=1}^{T} \|g_t\|_2^2 \leq \frac{D^2}{2\eta_\theta} + \frac{\eta_\theta}{2} \sum_{t=1}^{T} \|g_t\|_2^2.$$

Taking conditional expectation and using $\mathbb{E}[g_t \mid \theta_t, q_t] = \nabla_\theta f(\theta_t, q_t)$, convexity of $f(\cdot, q_t)$, and the conditional second-moment bound $\mathbb{E}[\|g_t\|_2^2 \mid \theta_t, q_t] \leq G_{\text{sg}}^2$ gives

$$\mathbb{E}\left[ \sum_{t=1}^{T} f(\theta_t, q_t) - f(\theta, q_t) \right] \leq \frac{D^2}{2\eta_\theta} + \frac{\eta_\theta G_{\text{sg}}^2 T}{2}.$$

Dividing by $T$ gives the learner's average regret bound.

**Step 2: adversary regret.** Let $u_t \in \mathbb{R}^B$ denote the (possibly estimated) payoff vector with entries $u_{t,b} \triangleq \hat{L}_{t,b}$. Exponentiated-gradient mirror ascent on the simplex with entropy regularizer satisfies the standard bound: for any $q \in \Delta_B$,

$$\sum_{t=1}^{T} \langle q, u_t \rangle - \langle q_t, u_t \rangle \leq \frac{\log B}{\eta_q} + \frac{\eta_q}{8} \sum_{t=1}^{T} \|u_t\|_\infty^2,$$

where the constant $1/8$ follows from Hoeffding's lemma when $u_{t,b} \in [0, M]$. Since $\|u_t\|_\infty \leq M$ a.s., we have

$$\mathbb{E}\left[\sum_{t=1}^T \langle q, \hat{L}_t \rangle - \langle q_t, \hat{L}_t \rangle\right] \leq \frac{\log B}{\eta_q} + \frac{\eta_q M^2 T}{8}.$$

Using unbiasedness $\mathbb{E}[\hat{L}_{t,b} \mid \theta_t] = L_b(\theta_t)$ yields the same bound with $L(\theta_t)$ in place of $\hat{L}_t$.

**Step 3: combine regrets into a saddle-point gap.** The learner regret implies

$$\frac{1}{T}\sum_{t=1}^T f(\theta_t, q_t) \leq \min_{\theta \in \Theta} \frac{1}{T}\sum_{t=1}^T f(\theta, q_t) + \frac{D^2}{2\eta_\theta T} + \frac{\eta_\theta G_{\text{sg}}^2}{2}.$$

The adversary regret implies

$$\max_{q \in \Delta_B} \frac{1}{T}\sum_{t=1}^T f(\theta_t, q) \leq \frac{1}{T}\sum_{t=1}^T f(\theta_t, q_t) + \frac{\log B}{\eta_q T} + \frac{\eta_q M^2}{8}.$$

Combining and using Jensen's inequality,

$$\max_{q \in \Delta_B} f(\bar{\theta}, q) \leq \max_{q \in \Delta_B} \frac{1}{T}\sum_{t=1}^T f(\theta_t, q) \leq \min_{\theta \in \Theta} \frac{1}{T}\sum_{t=1}^T f(\theta, q_t) + \frac{D^2}{2\eta_\theta T} + \frac{\eta_\theta G_{\text{sg}}^2}{2} + \frac{\log B}{\eta_q T} + \frac{\eta_q M^2}{8}.$$

Finally, convexity of $f(\theta, \cdot)$ in $q$ implies $\min_{\theta \in \Theta} f(\theta, \bar{q}) \leq \min_{\theta \in \Theta} \frac{1}{T}\sum_{t=1}^T f(\theta, q_t)$. Rearranging yields (33). The suboptimality bound (34) follows since $\max_{q \in \Delta_B} f(\bar{\theta}, q) = \max_b L_b(\bar{\theta})$. □

*Remark* E.8 (Reading Theorem E.7 in the deep RL regime). Assumption E.6 is not satisfied by neural policies trained with GRPO, so Theorem E.7 should be read as an *idealized online-learning lens* rather than a literal convergence guarantee. It formalizes the conceptual statement that if (i) the policy updates behave like a low-regret learner and (ii) the group reweighting behaves like a low-regret adversary (implemented via exponential weights), then the time averages approximate a GDRO saddle point.

Empirically, several qualitative predictions of this lens appear in our Prompt-GDRO dynamics: the adversary maintains a non-degenerate, entropy-regularized weight distribution over bins (see the entropy traces in Figure 4), and the weights form a "traveling wave frontier that leads the empirical prompt-share distribution early in training (visible in the triptych of Figure 2 and summarized by the lead–lag proxy in Figure 6a). In practice, Prompt-GDRO uses bandit feedback (we only observe losses for sampled prompts/bins), hence our implementation uses the GDRO-EXP3P estimator/updates of Soma et al. (2022), which preserve no-regret guarantees under partial information.

*Remark* E.9 (Where Rollout-GDRO enters the saddle-gap bound). The learner term $\frac{\eta_\theta}{2}\sum_{t=1}^T \|g_t\|_2^2$ in Theorem E.7 makes explicit that *estimation noise* impacts the constants in our no-regret analysis. In GRPO, $g_t$ is formed from Monte Carlo rollouts, and is therefore stochastic even when conditioning on $(\theta_t, q_t)$.

To separate optimization from estimation effects, write $g_t = \nabla_\theta f(\theta_t, q_t) + \xi_t$ with $\mathbb{E}[\xi_t \mid \theta_t, q_t] = 0$. Then

$$\mathbb{E}\|g_t\|_2^2 = \|\nabla_\theta f(\theta_t, q_t)\|_2^2 + \mathbb{E}\|\xi_t\|_2^2.$$

Now consider a batch of $M$ prompts whose (empirical) bin fractions are $\bar{q}_{t,1:B}$ and whose per-bin rollout allocation is $\mathbf{n}_t = (n_{t,1}, \ldots, n_{t,B})$. Under Lemma E.15, the stochastic component of the batch gradient obeys the proxy bound

$$\mathbb{E}\|\xi_t\|_2^2 \leq \frac{1}{M}\sum_{b=1}^B \bar{q}_{t,b} \frac{v_b(\theta_t)}{n_{t,b}},$$

where $v_b(\theta_t)$ is the intrinsic per-bin gradient-variance term. Rollout-GDRO is designed to *adaptively choose* $\mathbf{n}_t$ to reduce this variance proxy under a mean compute constraint. In Section E.2.4, we show that the rollout controller can itself be viewed as a no-regret primal–dual algorithm for a budgeted variance objective, making precise (in an idealized model) how variance-aware compute allocation tightens the $\sum_t \|g_t\|_2^2$ term relative to uniform rollouts.

### E.2. Rollout-GDRO as Variance-Aware Allocation Under a Budget and No-Regret Game Dynamics

We now analyze the second controller: allocating the number of rollouts per prompt as a function of group/bin. The key idea is classical: if some bins induce higher intrinsic variance (more stochastic rewards / longer reasoning / more fragile completions), then allocating *more* rollouts to these bins reduces gradient variance and improves stability.

#### E.2.1. A VARIANCE PROXY FOR GRPO ROLLOUTS

**Reference equations from the main paper.** In the main paper (Section 3), Rollout-GDRO is posed as a mean-rollout-budget allocation problem driven by empirical bin utilities. For completeness (and to resolve cross-references after removing the Method section), we restate the defining equations here.

$$\hat{J}_b(\theta; n_b) \triangleq -\frac{1}{|\mathcal{B}_{b,t}|} \sum_{x_i \in \mathcal{B}_{b,t}} \ell(x_i; \theta, n_b),  \tag{35}$$

where $\ell(x_i; \theta, n_b)$ denotes the prompt-level GRPO loss estimate based on $n_b$ rollouts.

$$\max_{\{n_b\}} \sum_{b=1}^{B} \hat{q}_t(b)\, \hat{J}_b(\theta; n_b) \quad \text{s.t.} \quad \sum_{b=1}^{B} \hat{q}_t(b)\, n_b = \bar{n}.  \tag{36}$$

$$L_b(n) = -\hat{J}_b(\theta; n) + \mu\, n.  \tag{37}$$

$$\mu \leftarrow \mu + \alpha_\mu(\hat{n}_{\text{realized}} - \bar{n}).  \tag{38}$$

At a training step $t$, the allocator observes the realized bin fractions $\hat{q}_t \in \Delta_B$ in the current prompt batch and chooses discrete rollout counts $n_b \in \{n_{\min}, \dots, n_{\max}\}$ for each bin $b$ under the mean rollout budget constraint in (36). The bin-level quantity $\hat{J}_b(\theta; n_b)$ in (35) is itself estimated from $n_b$ rollouts per prompt, so changing $n_b$ affects both the quality (variance) and possibly the bias of the signal provided to the learner.

**What is guaranteed by the rollout controller vs. what is explained by the variance proxy.** Equation (36) is the *implemented* allocation problem: at each step, the controller selects discrete arms $\{n_b\}$ to maximize an empirical utility under a strict mean-rollout constraint. This can be cast as an online constrained bandit problem by defining an (arm) loss $V_b(n; \theta) \triangleq -\hat{J}_b(\theta; n)$ (or any bounded monotone transform thereof), and augmenting it with the linear budget penalty $\mu n$ in (37). Our no-regret result in Section E.2.4 applies to this generic primal–dual EXP3P controller for (36). The variance-proxy analysis below is a *separate* guarantee: it upper-bounds the conditional variance of GRPO's Monte Carlo gradient estimator as a function of the allocation $\mathbf{n}$, yielding the proxy objective $\sum_b \bar{q}_b v_b(\theta)/n_b$ (Eq. (20)). In particular, if the controller is instantiated with feedback that estimates (minus) this proxy cost, then the same no-regret machinery yields a provably near-optimal variance-minimizing allocation.

Our theoretical results in this subsection focus on the stabilizing effect of increasing $n_b$ through variance reduction. Formally, we derive a simple proxy for the stochastic component of the GRPO batch gradient as a function of the rollout allocation $\mathbf{n} = (n_1, \dots, n_B)$, which yields an optimization problem of the form $\min_{\mathbf{n}} \sum_b \bar{q}_b v_b(\theta)/n_b$ under the same mean budget. Informally, the bin-dependent quantity $v_b(\theta)$ measures how noisy the per-rollout GRPO gradient is within bin $b$: larger $v_b(\theta)$ means completions in that bin yield higher-variance gradients, so averaging more rollouts (larger $n_b$) is more valuable there. For the cleanest bound, we define $v_b(\theta)$ as a uniform (worst-case) upper bound over prompts in bin $b$; see Lemma E.15. This makes explicit how Rollout-GDRO can be interpreted as variance-aware compute redistribution in the sense visualized by our "weighted standard error" diagnostics. For clarity, we fix a training step and write $\bar{q}_b \triangleq \hat{q}_t(b)$ for the realized bin fractions. That is, for the step-$t$ prompt batch $\{x_i\}_{i=1}^{M}$, $\hat{q}_t(b) \triangleq \frac{1}{M} \sum_{i=1}^{M} \mathbf{1}\{g(x_i) = b\}$, so $\bar{q}_b$ is the empirical fraction of bin-$b$ prompts in the batch.

Fix a prompt $x$ and policy parameters $\theta$. Let $\{y_j\}_{j=1}^{n_b}$ denote $n_b$ rollouts sampled from $\pi_\theta(\cdot \mid x)$. In GRPO, these rollouts are used to form a *prompt-level empirical loss* (cf. Section 2.2), which may involve within-prompt normalization across

the rollout group (e.g., the standardized advantages in (3)). We write this generic prompt-level estimator as $\hat{\ell}(x; \theta, n_b)$ and define the corresponding per-prompt gradient estimator

$$\hat{g}(x; \theta, n_b) \triangleq \nabla_\theta \hat{\ell}(x; \theta, n_b), \qquad g(x) = b. \tag{39}$$

The analysis below only requires that the rollouts are i.i.d. and that $\hat{g}$ is a (possibly coupled) function of these rollouts.

**Assumption E.10** (Rollout sampling model and differentiation under the expectation). Conditional on a prompt $x$ and parameters $\theta$, the rollouts $y_1, \ldots, y_{n_b}$ are drawn i.i.d. from $\pi_\theta(\cdot \mid x)$. Moreover, the prompt-level estimator $\hat{\ell}(x; \theta, n_b)$ is differentiable in $\theta$ and we may interchange gradient and conditional expectation: $\nabla_\theta \mathbb{E}[\hat{\ell}(x; \theta, n_b) \mid x] = \mathbb{E}[\nabla_\theta \hat{\ell}(x; \theta, n_b) \mid x]$.

**Lemma E.11** (Connecting $\hat{J}_b(\theta; n_b)$ to per-prompt gradient estimators). *Recall that $\hat{\ell}(x; \theta, n_b)$ denotes the prompt-level empirical loss estimator computed from $n_b$ rollouts, and $\hat{g}(x; \theta, n_b) = \nabla_\theta \hat{\ell}(x; \theta, n_b)$. If a bin-level estimator $\hat{J}_b(\theta; n_b)$ (as in (35)) is formed by averaging $\hat{\ell}(x; \theta, n_b)$ over the prompts $x$ in bin $b$ in the current batch (up to the sign convention $L_b = -J_b$), then $\nabla_\theta \hat{J}_b(\theta; n_b)$ is the corresponding average of the per-prompt estimators $\hat{g}(x; \theta, n_b)$.*

*Proof.* Both claims follow immediately from linearity of averaging and differentiation. $\square$

**Assumption E.12** (Bounded differences for the prompt-gradient estimator). For any prompt $x$ with $g(x) = b$ and any rollout count $n_b$, the estimator $\hat{g}(x; \theta, n_b)$ is a measurable function of the rollout group $(y_1, \ldots, y_{n_b})$. Assume there exists a (bin-dependent) constant $C_b(\theta) \geq 0$ such that, for every coordinate $j \in \{1, \ldots, n_b\}$ and any replacement rollout $y_j'$, the estimator satisfies the bounded-differences property

$$\left\| \hat{g}(x; \theta, n_b; y_1, \ldots, y_j, \ldots, y_{n_b}) - \hat{g}(x; \theta, n_b; y_1, \ldots, y_j', \ldots, y_{n_b}) \right\|_2 \leq \frac{C_b(\theta)}{n_b}. \tag{40}$$

**Lemma E.13** (GRPO prompt-gradient satisfies bounded differences under within-group normalization). *Consider a fixed prompt $x$ in bin $b$ and $n_b$ i.i.d. rollouts $y_1, \ldots, y_{n_b} \sim \pi_\theta(\cdot \mid x)$ with bounded rewards $r_j \triangleq r(x, y_j) \in [0, R]$. Let $\bar{r} \triangleq \frac{1}{n_b} \sum_{j=1}^{n_b} r_j$, $s \triangleq \sqrt{\frac{1}{n_b} \sum_{j=1}^{n_b} (r_j - \bar{r})^2}$, and define standardized advantages $A_j \triangleq (r_j - \bar{r})/(s + \varepsilon)$ for some fixed $\varepsilon > 0$. Suppose the (per-rollout) score function is uniformly bounded, i.e.,*

$$\left\| \nabla_\theta \log \pi_\theta(y \mid x) \right\|_2 \leq G_\pi \qquad \text{for all } (x, y, \theta). \tag{41}$$

*Consider the normalized prompt-gradient estimator*

$$\hat{g}(x; \theta, n_b) \triangleq \frac{1}{n_b} \sum_{j=1}^{n_b} A_j \nabla_\theta \log \pi_\theta(y_j \mid x). \tag{42}$$

*Then Assumption E.12 holds with a constant of the form*

$$C_b(\theta) \leq G_\pi \left( \frac{3R^2}{\varepsilon^2} + \frac{5R}{\varepsilon} \right). \tag{43}$$

*The same conclusion holds (up to replacing $G_\pi$ by an appropriate bound on the per-rollout GRPO score term) when additional bounded multiplicative factors are present, such as PPO ratio clipping.*

*Proof.* Fix an index $k \in \{1, \ldots, n_b\}$ and let $r = (r_1, \ldots, r_{n_b})$ and $r'$ denote the reward vectors that differ only at coordinate $k$ (corresponding to replacing $y_k$ by some alternative rollout $y_k'$). Let $(\bar{r}, s)$ and $(\bar{r}', s')$ denote the corresponding means and standard deviations, and define $A_j$ and $A_j'$ from $r$ and $r'$.

First, the mean changes by at most

$$|\bar{r} - \bar{r}'| \leq \frac{|r_k - r_k'|}{n_b} \leq \frac{R}{n_b}. \tag{44}$$

Next, writing the (biased) sample variance as $v \triangleq s^2 = \frac{1}{n_b}\sum_j r_j^2 - \bar{r}^2$ (and similarly $v' \triangleq s'^2$; biased vs. unbiased only changes constants), we have

$$
\begin{aligned}
|v - v'| &\leq \frac{1}{n_b}|r_k^2 - r_k'^2| + |\bar{r}^2 - \bar{r}'^2| \leq \frac{R^2}{n_b} + |\bar{r} - \bar{r}'|\,|\bar{r} + \bar{r}'| \\
&\leq \frac{R^2}{n_b} + \frac{R}{n_b}\cdot 2R \leq \frac{3R^2}{n_b}.
\end{aligned}
\tag{45}
$$

By the identity $|\sqrt{v} - \sqrt{v'}| = |v - v'|/(s + s')$ (interpreting the right-hand side as $0$ when $s = s' = 0$),

$$
|s - s'| \leq \frac{|v - v'|}{s + s'} \leq \frac{3R^2}{n_b\,(s + s')}.
\tag{46}
$$

Now decompose the change in the estimator (42) as

$$
\|\hat{g} - \hat{g}'\|_2 \leq \frac{1}{n_b}\sum_{j=1}^{n_b}\big\|(A_j - A_j')\nabla_\theta \log \pi_\theta(y_j \mid x)\big\|_2 \;+\; \frac{1}{n_b}\big\|A_k'\big(\nabla_\theta \log \pi_\theta(y_k \mid x) - \nabla_\theta \log \pi_\theta(y_k' \mid x)\big)\big\|_2.
\tag{47}
$$

Using (41) and $|A_k'| \leq \frac{|r_k' - \bar{r}'|}{\varepsilon} \leq \frac{R}{\varepsilon}$, the second term in (47) is at most $\frac{2RG_\pi}{\varepsilon n_b}$.

For the first term, we bound the total change in standardized advantages. For any $j \neq k$ (so $r_j = r_j'$), we can write

$$
\begin{aligned}
|A_j - A_j'| &= \left|\frac{r_j - \bar{r}}{s + \varepsilon} - \frac{r_j - \bar{r}'}{s' + \varepsilon}\right| \\
&\leq \frac{|\bar{r} - \bar{r}'|}{s + \varepsilon} \;+\; |r_j - \bar{r}|\left|\frac{1}{s + \varepsilon} - \frac{1}{s' + \varepsilon}\right|.
\end{aligned}
\tag{48}
$$

Summing (48) over $j \neq k$ and using (44) gives

$$
\sum_{j\neq k}\frac{|\bar{r} - \bar{r}'|}{s + \varepsilon} \leq (n_b - 1)\cdot\frac{R/n_b}{\varepsilon} \leq \frac{R}{\varepsilon}.
\tag{49}
$$

For the denominator term, note that $\left|\frac{1}{s+\varepsilon} - \frac{1}{s'+\varepsilon}\right| = \frac{|s-s'|}{(s+\varepsilon)(s'+\varepsilon)} \leq \frac{|s-s'|}{\varepsilon^2}$. Also, by Cauchy–Schwarz, $\sum_{j=1}^{n_b}|r_j - \bar{r}| \leq \sqrt{n_b\sum_j(r_j - \bar{r})^2} = n_b s$. Combining with (46) and (45) yields

$$
\sum_{j\neq k}|r_j - \bar{r}|\left|\frac{1}{s+\varepsilon} - \frac{1}{s'+\varepsilon}\right| \leq \frac{n_b s}{\varepsilon^2}\cdot\frac{3R^2}{n_b(s+s')} \leq \frac{3R^2}{\varepsilon^2}.
\tag{50}
$$

Finally, using the crude bound $|A_k - A_k'| \leq |A_k| + |A_k'| \leq 2R/\varepsilon$ for the changed coordinate, we obtain

$$
\sum_{j=1}^{n_b}|A_j - A_j'| \leq \frac{3R^2}{\varepsilon^2} + \frac{3R}{\varepsilon}.
\tag{51}
$$

Plugging (51) into the first term of (47) and combining with the second-term bound gives $\|\hat{g} - \hat{g}'\|_2 \leq \frac{G_\pi}{n_b}\big(\frac{3R^2}{\varepsilon^2} + \frac{5R}{\varepsilon}\big)$, which is the stated bounded-differences form. $\qquad\square$

**Lemma E.14** (A $1/n$ variance bound (covers within-prompt normalization in GRPO)). *Assume Assumptions E.10 and E.12 hold. Then for any prompt $x$ with $g(x) = b$,*

$$
\mathbb{E}\left[\|\hat{g}(x;\theta,n_b) - \mathbb{E}[\hat{g}(x;\theta,n_b) \mid x]\|_2^2 \,\Big|\, x\right] \leq \frac{C_b(\theta)^2}{2n_b}.
\tag{52}
$$

*In particular, defining $v_b(\theta) \triangleq C_b(\theta)^2/2$ yields the convenient form $\mathbb{E}[\|\hat{g}(x;\theta,n_b) - \mathbb{E}[\hat{g}(x;\theta,n_b) \mid x]\|_2^2 \mid x] \leq v_b(\theta)/n_b$.*

*Proof.* Fix $x$ and write $Y = (y_1, \ldots, y_{n_b})$ for the rollout group. Let $Y^{(j)}$ denote the vector obtained by replacing only the $j$-th coordinate $y_j$ with an independent copy $y'_j \sim \pi_\theta(\cdot \mid x)$. *Fact (vector-valued Efron–Stein).* The Efron–Stein inequality extends to $\mathbb{R}^d$-valued estimators with $\| \cdot \|_2$ (more generally, Hilbert space-valued) by applying the scalar inequality coordinatewise and summing; see, e.g., Boucheron et al. (2013, Ch. 3). The Efron–Stein inequality (vector-valued Efron–Stein / Hilbert space version) implies

$$\mathbb{E}\left[\|\hat{g}(Y) - \mathbb{E}[\hat{g}(Y) \mid x]\|_2^2 \,\Big|\, x\right] \;\leq\; \frac{1}{2}\sum_{j=1}^{n_b} \mathbb{E}\left[\left\|\hat{g}(Y) - \hat{g}\left(Y^{(j)}\right)\right\|_2^2 \,\Big|\, x\right].$$

By Equation (40), each summand is at most $C_b(\theta)^2/n_b^2$. Summing over $j$ gives (52). $\qquad\square$

Next, consider a batch of $M$ prompts $\{x_i\}_{i=1}^M$. Let $\mathbf{n} = (n_1, \ldots, n_B)$ denote the vector of per-bin rollout counts, and define the batch gradient estimator

$$\hat{g}(\theta; \mathbf{n}) \;\triangleq\; \frac{1}{M}\sum_{i=1}^M \hat{g}(x_i; \theta, n_{g(x_i)}). \tag{53}$$

**Lemma E.15** (A variance proxy decomposes over groups)**.** *Assume that, conditioned on the batch prompts $\{x_i\}_{i=1}^M$, the rollout groups used to form the per-prompt estimators $\hat{g}(x_i; \theta, n_{g(x_i)})$ are independent across $i$. Fix a batch of $M$ prompts $\{x_i\}_{i=1}^M$ and define the empirical bin fractions $\bar{q}_b \triangleq \frac{1}{M}\sum_{i=1}^M \mathbf{1}\{g(x_i) = b\}$. Then, conditioned on the batch prompts $\{x_i\}_{i=1}^M$,*

$$\mathbb{E}\left[\|\hat{g}(\theta; \mathbf{n}) - \mathbb{E}[\hat{g}(\theta; \mathbf{n}) \mid \{x_i\}_{i=1}^M]\|_2^2 \,\Big|\, \{x_i\}_{i=1}^M\right] \;\leq\; \frac{1}{M}\sum_{b=1}^B \bar{q}_b \frac{v_b(\theta)}{n_b}, \tag{54}$$

*where $v_b(\theta)$ is any uniform bin-wise constant such that the per-prompt conditional variance obeys $\mathbb{E}[\|\hat{g}(x; \theta, n_b) - \mathbb{E}[\hat{g}(x; \theta, n_b) \mid x]\|_2^2 \mid x] \leq v_b(\theta)/n_b$ for all prompts $x$ with $g(x) = b$. Under Assumption E.12 and Lemma E.14, we may take $v_b(\theta) = C_b(\theta)^2/2$.*

*Proof.* Write $\hat{g}(\theta; \mathbf{n}) - \mathbb{E}[\hat{g}(\theta; \mathbf{n}) \mid \{x_i\}_{i=1}^M] = \frac{1}{M}\sum_{i=1}^M Z_i$, where $Z_i \triangleq \hat{g}(x_i; \theta, n_{g(x_i)}) - \mathbb{E}[\hat{g}(x_i; \theta, n_{g(x_i)}) \mid x_i]$. Conditioned on the prompts, $\{Z_i\}_{i=1}^M$ are independent and zero-mean. Thus,

$$\mathbb{E}\left[\left\|\frac{1}{M}\sum_{i=1}^M Z_i\right\|_2^2 \,\Big|\, \{x_i\}_{i=1}^M\right] = \frac{1}{M^2}\sum_{i=1}^M \mathbb{E}[\|Z_i\|_2^2 \mid x_i] \leq \frac{1}{M^2}\sum_{i=1}^M \frac{v_{g(x_i)}(\theta)}{n_{g(x_i)}},$$

where the last inequality uses Lemma E.14 and the definition of $v_b(\theta)$. Grouping terms by bins yields $\frac{1}{M^2}\sum_{b=1}^B \sum_{i:g(x_i)=b} \frac{v_b(\theta)}{n_b} = \frac{1}{M}\sum_{b=1}^B \bar{q}_b \frac{v_b(\theta)}{n_b}$, which proves (54). $\qquad\square$

*Remark* E.16 (From variance proxy to the plotted "weighted standard error")**.** The quantity on the right-hand side of (54) is a *variance proxy*. In experiments we additionally visualize the more interpretable "weighted standard error" proxy $\sum_b \bar{q}_b \sqrt{v_b(\theta)/n_b}$. This proxy is often easier to interpret because each term $\sqrt{v_b(\theta)/n_b}$ behaves like a standard error: it is a within-bin scale of the prompt-gradient noise, which shrinks at the canonical $1/\sqrt{n_b}$ rate when using $n_b$ rollouts (Lemma E.14), and the weights $\bar{q}_b$ simply average these standard-error contributions across the observed batch composition (see Figure 6b). By Cauchy–Schwarz,

$$\left(\sum_{b=1}^B \bar{q}_b \sqrt{\frac{v_b(\theta)}{n_b}}\right)^2 \leq \left(\sum_{b=1}^B \bar{q}_b\right)\left(\sum_{b=1}^B \bar{q}_b \frac{v_b(\theta)}{n_b}\right) = \sum_{b=1}^B \bar{q}_b \frac{v_b(\theta)}{n_b}.$$

Thus, minimizing the variance proxy also controls the weighted standard-error proxy we plot.

E.2.2. THE VARIANCE-OPTIMAL ALLOCATION OBEYS A SQUARE-ROOT LAW

Lemma E.15 suggests that, under a fixed mean rollout budget, the rollout allocation $\mathbf{n}$ controls a leading-order proxy for the stochastic component of the learner's update through the term $\sum_b \bar{q}_b \, v_b(\theta)/n_b$. This motivates studying a variance-aware relaxation of the allocator objective (35), in which the *marginal value* of additional rollouts is captured by the reduction in estimator variance. Concretely, we fix $\theta$ and treat $v_b(\theta)$ and $\bar{q}_b$ as constants for a given step (with $\bar{q}_b = \hat{q}_t(b)$), and we analyze the continuous relaxation

$$\min_{\mathbf{n} \in \mathbb{R}_+^B} \sum_{b=1}^{B} \bar{q}_b \, \frac{v_b(\theta)}{n_b} \qquad \text{s.t.} \qquad \sum_{b=1}^{B} \bar{q}_b \, n_b = \bar{n}. \tag{55}$$

This objective is convex in $\mathbf{n}$ (each term is $1/n_b$), and the constraint is linear.

**Theorem E.17** (Square-root law for variance-optimal allocation). *Let $A \triangleq \{b \in [B] : \bar{q}_b > 0\}$ denote the set of bins present in the batch. If $v_b(\theta) > 0$ for all $b \in A$, then the minimizer of (55) is unique on the active coordinates and is*

$$n_b^\star = \bar{n} \cdot \frac{\sqrt{v_b(\theta)}}{\sum_{j=1}^{B} \bar{q}_j \, \sqrt{v_j(\theta)}}, \qquad b \in A. \tag{56}$$

*For bins with $\bar{q}_b = 0$ (i.e., $b \notin A$), the variable $n_b$ does not affect the objective nor the budget constraint and may be chosen arbitrarily.*

*Moreover, the minimal objective value equals*

$$\sum_{b=1}^{B} \bar{q}_b \, \frac{v_b(\theta)}{n_b^\star} = \frac{\left( \sum_{b=1}^{B} \bar{q}_b \, \sqrt{v_b(\theta)} \right)^2}{\bar{n}}. \tag{57}$$

*Compared to the uniform allocation $n_b \equiv \bar{n}$, the optimal allocation never increases the variance proxy:*

$$\sum_{b=1}^{B} \bar{q}_b \, \frac{v_b(\theta)}{n_b^\star} \; \leq \; \sum_{b=1}^{B} \bar{q}_b \, \frac{v_b(\theta)}{\bar{n}} = \frac{\sum_{b=1}^{B} \bar{q}_b v_b(\theta)}{\bar{n}}, \tag{58}$$

*with equality iff $v_b(\theta)$ is constant across the active bins $b \in A$.*

*Proof.* If $v_b(\theta) = 0$ for some active bin $b \in A$, then its term $\bar{q}_b v_b(\theta)/n_b$ is identically zero, so allocating additional rollouts to that bin cannot reduce the variance proxy; in that degenerate case one may treat such bins as "free" and apply the same square-root allocation to the remaining bins with positive $v_b(\theta)$ (in our discrete implementation, this corresponds to choosing the smallest rollout arm). Form the Lagrangian (with multiplier $\mu$, matching the "shadow price" interpretation in (36))

$$\mathcal{L}(\mathbf{n}, \mu) = \sum_{b=1}^{B} \bar{q}_b \frac{v_b(\theta)}{n_b} + \mu \left( \sum_{b=1}^{B} \bar{q}_b n_b - \bar{n} \right).$$

Bins with $\bar{q}_b = 0$ do not appear in the objective nor the constraint in (55), so we may ignore them and restrict attention to the active set $A$. Stationarity gives, for each active bin $b \in A$ (so $\bar{q}_b > 0$),

$$-\bar{q}_b \frac{v_b(\theta)}{n_b^2} + \mu \bar{q}_b = 0 \quad \implies \quad n_b = \sqrt{\frac{v_b(\theta)}{\mu}}.$$

Enforcing the constraint yields $\sum_b \bar{q}_b \sqrt{v_b(\theta)/\mu} = \bar{n}$, hence $\sqrt{\mu} = \frac{\sum_j \bar{q}_j \sqrt{v_j(\theta)}}{\bar{n}}$, and substituting gives (56). Uniqueness on the active coordinates follows because the objective in (55) is strictly convex in $\{n_b\}_{b \in A}$ when $v_b(\theta) > 0$.

Plugging (56) into the objective yields (57).

Finally, to show (58), apply Cauchy–Schwarz:

$$\left( \sum_{b=1}^{B} \bar{q}_b \sqrt{v_b(\theta)} \right)^2 \leq \left( \sum_{b=1}^{B} \bar{q}_b \right) \left( \sum_{b=1}^{B} \bar{q}_b v_b(\theta) \right) = \sum_{b=1}^{B} \bar{q}_b v_b(\theta).$$

Divide by $\bar{n}$ and use (57) to obtain (58). Equality holds iff $\sqrt{v_b(\theta)}$ is constant across $b \in A$. $\qquad \square$

*Remark* E.18 (Interpretation). Theorem E.17 shows that the variance-optimal allocation equalizes the *marginal* variance proxy across bins: at the optimum, $v_b(\theta)/n_b^{\star 2} = \mu$ is constant in $b$ (the KKT multiplier / shadow price). Consequently, bins with larger intrinsic variance $v_b(\theta)$ receive more rollouts, with *per-prompt* rollouts scaling as $n_b^\star \propto \sqrt{v_b(\theta)}$. This square-root law is the same structure as the classical Neyman allocation in stratified sampling and Monte Carlo budgeting (see, e.g., Rubinstein & Kroese, 2016). Notably, the bin fraction $\bar{q}_b$ cancels from the stationarity condition, meaning that the *marginal* value of extra rollouts depends on $v_b(\theta)$ rather than frequency. At the same time, the budget is weighted by $\bar{q}_b$, so rare bins can receive large $n_b^\star$ without violating the mean rollout budget $\sum_b \bar{q}_b n_b = \bar{n}$. See the rollout heatmaps and time snapshots in Figures 3 and 8 for the corresponding piecewise-constant ("staircase") transitions between discrete rollout arms as the shadow price $\mu$ adapts. This square-root allocation intuition is consistent with the empirically observed rollout-allocation patterns in Rollout-GDRO (see, e.g., Figures 3 and 8), where bins deemed noisier receive larger rollout arms under a fixed mean budget.

### E.2.3. DISCRETE ROLLOUT ARMS AND AN ENTROPIC PRIMAL–DUAL VIEW

The continuous analysis in Section E.2.1–Section E.2.2 yields a closed-form allocation for the idealized variance proxy objective when the rollout counts $\{n_b\}$ are allowed to be any positive reals. Rollout-GDRO, however, must choose *discrete* rollout counts $n_b \in \mathcal{N} = \{n_{\min}, \dots, n_{\max}\}$ and enforce the mean rollout budget online. Mirroring the Lagrangian perspective in the main paper (cf. (36)), we view the allocator as maintaining a dual variable $\mu$ that acts as a *shadow price of compute*. To connect the discrete implementation to the variance proxy derived above, we consider an entropy-regularized relaxation of the same constrained problem.

Concretely, for each bin $b$ we maintain a distribution $p_b \in \Delta_{\mathcal{N}}$ over rollout "arms" $n \in \mathcal{N}$. We model the per-bin *cost* of choosing arm $n$ as a function $V_b(n; \theta)$ that decreases with $n$. In the variance-proxy setting of Section E.2.1, a canonical choice is $V_b(n; \theta) = v_b(\theta)/n$. (Equivalently, one may interpret the utility in (35) as a monotone transform of $-V_b$; our analysis isolates the variance-reduction component that is most directly controlled by $n$.) The entropy-regularized Lagrangian for a fixed $\theta$ takes the form

$$\mathcal{L}(\{p_b\}, \mu; \theta) = \sum_{b=1}^{B} \bar{q}_b \left( \mathbb{E}_{n \sim p_b}[V_b(n; \theta)] - \frac{1}{\eta} H(p_b) \right) + \mu \left( \sum_{b=1}^{B} \bar{q}_b \, \mathbb{E}_{n \sim p_b}[n] - \bar{n} \right), \tag{59}$$

where $\mu \geq 0$ is the shadow price and the entropy term encourages exploration over arms (stability).

**Lemma E.19** (Soft-min solution for rollout arms). *For fixed $\mu$ and bin $b$, the minimizer of (59) over $p_b \in \Delta_{\mathcal{N}}$ is*

$$p_{b,\mu}^\star(n) = \frac{\exp\left( -\eta \left[ V_b(n; \theta) + \mu n \right] \right)}{\sum_{n' \in \mathcal{N}} \exp\left( -\eta \left[ V_b(n'; \theta) + \mu n' \right] \right)}. \tag{60}$$

*Moreover, the optimal value of the inner minimization equals a soft-min:*

$$\min_{p_b \in \Delta_{\mathcal{N}}} \left\{ \mathbb{E}_{n \sim p_b}[V_b(n; \theta) + \mu n] - \frac{1}{\eta} H(p_b) \right\} = -\frac{1}{\eta} \log \left( \sum_{n \in \mathcal{N}} e^{-\eta(V_b(n;\theta)+\mu n)} \right). \tag{61}$$

*Proof.* Apply Lemma E.1 to the vector $v \in \mathbb{R}^{|\mathcal{N}|}$ with entries $v_n \triangleq -\left( V_b(n; \theta) + \mu n \right)$. Then

$$\max_{p \in \Delta_{\mathcal{N}}} \left\{ \langle p, v \rangle + \frac{1}{\eta} H(p) \right\} = \frac{1}{\eta} \log \sum_{n \in \mathcal{N}} e^{\eta v_n} = \frac{1}{\eta} \log \sum_{n \in \mathcal{N}} e^{-\eta(V_b(n;\theta)+\mu n)}.$$

Negating both sides turns the maximization into the minimization in (61). The maximizer in Lemma E.1 becomes the minimizer here and equals (60). $\square$

**Corollary E.20** (Soft-min approximation quality). *Let $\mathrm{smin}_\eta(z) \triangleq -\frac{1}{\eta} \log \sum_{i=1}^{K} e^{-\eta z_i}$. Then for any $z \in \mathbb{R}^K$,*

$$\mathrm{smin}_\eta(z) \leq \min_i z_i \leq \mathrm{smin}_\eta(z) + \frac{\log K}{\eta}. \tag{62}$$

*Proof.* Apply Corollary E.2 to $v = -z$. $\square$

*Remark* E.21 (Shadow price and the "staircase" compute pattern). Lemma E.19 makes explicit that $\mu$ acts as a *shadow price* on rollouts: increasing $\mu$ shifts probability mass in (60) toward smaller rollout counts $n$. In our implementation, $\mu$ is updated by (projected) dual ascent on the budget violation, so the system dynamically finds a price at which the expected rollout budget is approximately satisfied. This primal–dual viewpoint also clarifies the connection to the square-root law. If we set $V_b(n;\theta) = v_b(\theta)/n$ and temporarily relax $n$ to be continuous, then the unregularized best response to a fixed price $\mu$ solves $\min_{n>0} v_b(\theta)/n + \mu n$, whose minimizer is $n_b(\mu) = \sqrt{v_b(\theta)/\mu}$. Choosing $\mu$ to satisfy the mean-budget constraint recovers the closed-form allocation in Theorem E.17. With a finite discrete arm set $\mathcal{N}$, the minimizing arm is the nearest available rollout count to $\sqrt{v_b(\theta)/\mu}$, so as $\mu$ changes the optimizer can switch abruptly between arms. This explains the "staircase" patterns observed in the rollout-allocation figures. See Figures 3 and 8 for these staircase-like transitions between discrete rollout arms as the learned shadow price $\mu$ changes.

### E.2.4. NO-REGRET PRIMAL–DUAL ANALYSIS FOR ROLLOUT-GDRO

The previous subsection gave a (regularized) Lagrangian view of the rollout controller. Here we show that the same structure admits a clean *no-regret* interpretation, mirroring the Prompt-GDRO analysis in Section E.1.

Throughout this subsection, we idealize the rollout controller as operating on a *fixed* rollout-cost landscape, i.e., we condition on a fixed policy parameter $\theta$ and fixed group fractions $\bar{q} \in \Delta_B$. This subsection analyzes Rollout-GDRO as an independent online budget-allocation game for fixed $(\theta, \bar{q})$. (We comment on the coupled, multi-time-scale setting in Remark E.24.)

**A truncated Lagrangian game.** Let $\mathcal{N} = \{n_{\min}, \ldots, n_{\max}\}$ be the discrete set of rollout arms and denote $K \triangleq |\mathcal{N}|$. For each bin $b$, the rollout controller chooses a distribution $p_b \in \Delta_{\mathcal{N}}$ over arms; write $p = (p_1, \ldots, p_B)$. Let $V_b(n;\theta) \geq 0$ denote the (idealized) variance proxy for choosing $n$ rollouts in bin $b$ at policy $\theta$ (e.g., $V_b(n;\theta) = v_b(\theta)/n$ as in Equation (55)). Let $a(n) \triangleq n$ denote the compute cost of arm $n$. We consider the convex–concave "truncated" Lagrangian

$$L_{\text{roll}}(p, \mu; \theta) \triangleq \sum_{b=1}^{B} \bar{q}_b \, \mathbb{E}_{n \sim p_b}[V_b(n;\theta)] \; + \; \mu \left( \sum_{b=1}^{B} \bar{q}_b \, \mathbb{E}_{n \sim p_b}[a(n)] - \bar{n} \right), \qquad \mu \in [0, \mu_{\max}], \tag{63}$$

where $\mu$ is the shadow price and $\mu_{\max}$ is a chosen truncation level (projection radius). When $\mu_{\max}$ is large, maximizing over $\mu$ approximately enforces the budget constraint.

**Primal–dual updates.** A natural no-regret dynamics for (63) is:

$$p_{t+1,b} = \arg\min_{p_b \in \Delta_{\mathcal{N}}} \left\{ \langle p_b, \ell_{t,b} \rangle + \frac{1}{\eta_p} \text{KL}(p_b \| p_{t,b}) \right\}, \qquad \ell_{t,b}(n) \triangleq \bar{q}_b \big( V_b(n;\theta) + \mu_t a(n) \big), \tag{64}$$

$$\mu_{t+1} = \Pi_{[0,\mu_{\max}]} \left( \mu_t + \eta_\mu \, g_t \right), \qquad g_t \triangleq \sum_{b=1}^{B} \bar{q}_b \, \mathbb{E}_{n \sim p_{t,b}}[a(n)] - \bar{n}, \tag{65}$$

where $\Pi_{[0,\mu_{\max}]}$ is Euclidean projection and $\text{KL}(\cdot \| \cdot)$ is the KL divergence. The update (64) is entropic mirror descent (a.k.a. exponentiated gradient) on the simplex, applied independently in each bin. Lemma E.19 shows that for fixed $\mu$, the minimizer has a Gibbs / soft-min form; the dynamics (64) tracks this solution online as $\mu_t$ evolves.

**Theorem E.22** (No-regret guarantee for Rollout-GDRO's primal–dual controller). *Assume that for all $b$ and $n \in \mathcal{N}$ we have $0 \leq V_b(n;\theta) \leq V_{\max}$ and $0 \leq a(n) \leq a_{\max}$. Let $K = |\mathcal{N}|$ and initialize $p_{1,b}$ to the uniform distribution on $\mathcal{N}$ and $\mu_1 = 0$. Run the updates (64)–(65) for $T$ steps with step sizes $\eta_p, \eta_\mu > 0$.*

*Define the averaged iterates $\bar{p}_b \triangleq \frac{1}{T} \sum_{t=1}^{T} p_{t,b}$ and $\bar{\mu} \triangleq \frac{1}{T} \sum_{t=1}^{T} \mu_t$. Then the (truncated) saddle-point gap satisfies*

$$\max_{\mu \in [0,\mu_{\max}]} L_{\text{roll}}(\bar{p}, \mu; \theta) \; - \; \min_{p \in (\Delta_{\mathcal{N}})^B} L_{\text{roll}}(p, \bar{\mu}; \theta) \; \leq \; \frac{B \log K}{\eta_p T} + \frac{\eta_p}{8} \left( V_{\max} + \mu_{\max} a_{\max} \right)^2 + \frac{\mu_{\max}^2}{2\eta_\mu T} + \frac{\eta_\mu}{2} a_{\max}^2. \tag{66}$$

*In particular, the explicit step sizes*

$$\eta_p \triangleq \frac{\sqrt{8B \log K}}{(V_{\max} + \mu_{\max} a_{\max})\sqrt{T}}, \qquad \eta_\mu \triangleq \frac{\mu_{\max}}{a_{\max}\sqrt{T}}, \tag{67}$$

*yield the concrete bound*

$$\mathrm{Gap}_T \;\le\; (V_{\max} + \mu_{\max} a_{\max}) \sqrt{\frac{B \log K}{2T}} \;+\; \frac{\mu_{\max} a_{\max}}{\sqrt{T}}. \tag{68}$$

*Moreover, let $p^\star$ be an optimal solution of the* budgeted variance *problem*

$$\min_{p \in (\Delta_{\mathcal{N}})^B} \sum_{b=1}^{B} \bar{q}_b \, \mathbb{E}_{n \sim p_b}[V_b(n;\theta)] \qquad s.t. \qquad \sum_{b=1}^{B} \bar{q}_b \, \mathbb{E}_{n \sim p_b}[a(n)] \le \bar{n}. \tag{69}$$

*Then the averaged rollout policy $\bar{p}$ is nearly optimal and nearly feasible:*

$$\sum_{b=1}^{B} \bar{q}_b \, \mathbb{E}_{n \sim \bar{p}_b}[V_b(n;\theta)] - \sum_{b=1}^{B} \bar{q}_b \, \mathbb{E}_{n \sim p_b^\star}[V_b(n;\theta)] \le \mathrm{Gap}_T, \tag{70}$$

$$\left[ \sum_{b=1}^{B} \bar{q}_b \, \mathbb{E}_{n \sim \bar{p}_b}[a(n)] - \bar{n} \right]_+ \le \frac{\sum_{b=1}^{B} \bar{q}_b \, \mathbb{E}_{n \sim p_b^\star}[V_b(n;\theta)] + \mathrm{Gap}_T}{\mu_{\max}} \le \frac{V_{\max} + \mathrm{Gap}_T}{\mu_{\max}}, \tag{71}$$

*where $\mathrm{Gap}_T$ denotes the right-hand side of* (66).

*Proof.* We apply a standard two-player no-regret-to-equilibrium argument to the convex–concave game (63); see, e.g., Cesa-Bianchi & Lugosi, 2006; Hazan, 2016; Bubeck, 2015 for the standard regret bounds and the conversion to saddle-point guarantees.

**Step 1: primal regret bound (entropic mirror descent).** Fix a bin $b$. The primal update (64) is entropic mirror descent on $\Delta_{\mathcal{N}}$ with loss vector $\ell_{t,b} \in \mathbb{R}^K$. Because $0 \le V_b(n;\theta) \le V_{\max}$, $0 \le a(n) \le a_{\max}$, and $0 \le \mu_t \le \mu_{\max}$, each coordinate satisfies $0 \le \ell_{t,b}(n) \le \bar{q}_b (V_{\max} + \mu_{\max} a_{\max})$. The standard entropic-mirror-descent regret bound (via the log-sum-exp potential and Hoeffding's lemma) gives, for any fixed comparator $p_b \in \Delta_{\mathcal{N}}$,

$$\sum_{t=1}^{T} \langle p_{t,b}, \ell_{t,b} \rangle - \sum_{t=1}^{T} \langle p_b, \ell_{t,b} \rangle \le \frac{\log K}{\eta_p} + \frac{\eta_p}{8} T \bar{q}_b^2 (V_{\max} + \mu_{\max} a_{\max})^2. \tag{72}$$

Summing (72) over $b = 1, \dots, B$ yields

$$\sum_{t=1}^{T} L_{\mathrm{roll}}(p_t, \mu_t; \theta) - \sum_{t=1}^{T} L_{\mathrm{roll}}(p, \mu_t; \theta) \le \frac{B \log K}{\eta_p} + \frac{\eta_p}{8} T (V_{\max} + \mu_{\max} a_{\max})^2 \sum_{b=1}^{B} \bar{q}_b^2 \tag{73}$$

$$\le \frac{B \log K}{\eta_p} + \frac{\eta_p}{8} T (V_{\max} + \mu_{\max} a_{\max})^2, \tag{74}$$

the last inequality holds due to the algebraic fact that $a^2 + \cdots + b^2 \le (a + \cdots + b)^2 \le 1$ for any probability measure $(a, \cdots, b) \in \Delta$, where $p = (p_1, \dots, p_B)$ and $p_t = (p_{t,1}, \dots, p_{t,B})$.

**Step 2: dual regret bound (projected gradient ascent).** The dual player maximizes $L_{\mathrm{roll}}(p_t, \mu; \theta)$ over $\mu \in [0, \mu_{\max}]$. Since $L_{\mathrm{roll}}(p_t, \mu; \theta)$ is linear in $\mu$, the dual gradient is exactly $g_t$ from (65). Moreover, because $0 \le a(n) \le a_{\max}$ and $\bar{q} \in \Delta_B$, we have $|g_t| \le a_{\max}$. The standard regret bound for projected online gradient ascent on an interval gives, for any fixed $\mu \in [0, \mu_{\max}]$,

$$\sum_{t=1}^{T} L_{\mathrm{roll}}(p_t, \mu; \theta) - \sum_{t=1}^{T} L_{\mathrm{roll}}(p_t, \mu_t; \theta) \le \frac{\mu_{\max}^2}{2\eta_\mu} + \frac{\eta_\mu}{2} T a_{\max}^2. \tag{75}$$

**Step 3: combine regrets and average.** Adding (73) and (75) and dividing by $T$ yields, for all $p$ and $\mu$,

$$\frac{1}{T} \sum_{t=1}^{T} L_{\mathrm{roll}}(p_t, \mu; \theta) - \frac{1}{T} \sum_{t=1}^{T} L_{\mathrm{roll}}(p, \mu_t; \theta) \le \mathrm{Gap}_T. \tag{76}$$

Because $L_{\mathrm{roll}}(\cdot, \mu; \theta)$ is convex in $p$ and $L_{\mathrm{roll}}(p, \cdot; \theta)$ is linear in $\mu$, Jensen's inequality gives

$$L_{\mathrm{roll}}(\bar{p}, \mu; \theta) \leq \frac{1}{T} \sum_{t=1}^{T} L_{\mathrm{roll}}(p_t, \mu; \theta), \qquad \frac{1}{T} \sum_{t=1}^{T} L_{\mathrm{roll}}(p, \mu_t; \theta) = L_{\mathrm{roll}}(p, \bar{\mu}; \theta).$$

Substitute into (76) to obtain (66).

**Step 4: deduce objective and budget bounds.** Let $p^\star$ be feasible for (69). For any $\mu \in [0, \mu_{\max}]$, feasibility implies $L_{\mathrm{roll}}(p^\star, \mu; \theta) \leq L_{\mathrm{roll}}(p^\star, 0; \theta)$. Thus

$$\min_p L_{\mathrm{roll}}(p, \bar{\mu}; \theta) \leq L_{\mathrm{roll}}(p^\star, \bar{\mu}; \theta) \leq L_{\mathrm{roll}}(p^\star, 0; \theta).$$

Combining with (66) yields
$$\max_{\mu \in [0, \mu_{\max}]} L_{\mathrm{roll}}(\bar{p}, \mu; \theta) \leq L_{\mathrm{roll}}(p^\star, 0; \theta) + \mathrm{Gap}_T.$$

Finally, observe that

$$\max_{\mu \in [0, \mu_{\max}]} L_{\mathrm{roll}}(\bar{p}, \mu; \theta) = \sum_{b=1}^{B} \bar{q}_b \, \mathbb{E}_{n \sim \bar{p}_b}[V_b(n; \theta)] \; + \; \mu_{\max} \left[ \sum_{b=1}^{B} \bar{q}_b \, \mathbb{E}_{n \sim \bar{p}_b}[a(n)] - \bar{n} \right]_+,$$

which immediately implies (70) and (71). $\qquad \square$

*Remark* E.23 (Bandit feedback and EXP3-style updates). Our implementation uses EXP3P-style updates because Rollout-GDRO only observes the variance proxy corresponding to the chosen arm. The full-information analysis above can be extended to the bandit setting by replacing $\ell_{t,b}$ with an unbiased importance-weighted estimator and adding the usual $\sqrt{K}$ factor in the regret term. We omit the (standard) bandit algebra here, since the qualitative conclusion remains the same: the rollout controller is a no-regret player in a budgeted Lagrangian game.

*Remark* E.24 (Decoupled analysis and possible coupling in the full system). See the Limitations and Future Work section for further discussion of this coupled setting and open problems it raises. In this paper we analyze and evaluate *Prompt-GDRO* and *Rollout-GDRO* as *separate* controllers. Accordingly, in the Rollout-GDRO analysis we treat the group fractions $\bar{q} \in \Delta_B$ as *exogenous* (a property of the current training data pipeline and grouping rule), and we study how the rollout allocator responds to group-dependent variance proxies under the mean-budget constraint $\sum_{b=1}^{B} \bar{q}_b n_b = \bar{n}$. If both controllers are enabled simultaneously, then $\bar{q}$ becomes endogenous to the Prompt-GDRO sampler and the training loop induces a coupled multi-time-scale game between the prompt sampler, the rollout allocator, and the learner. Characterizing stability and convergence of this coupled regime is left to future work.

*Remark* E.25 (Practical proxies for $v_b(\theta)$ in GRPO and connection to our implementation). The quantity $v_b(\theta)$ in Lemma E.15 is a group-dependent second-moment / variance proxy for a *single-prompt* stochastic gradient contribution. It enters the idealized variance-aware relaxation (55) only through the characteristic $1/n_b$ scaling obtained when using $n_b$ rollouts (Lemma E.14). In our Rollout-GDRO implementation (Section 3), each bin $b$ selects a discrete rollout arm $n \in \{n_{\min}, \ldots, n_{\max}\}$ using a GDRO-EXP3P bandit update driven by the augmented arm loss $\mathcal{L}_b(n) = \widehat{L}_b(\theta; n) + \mu\,(n - \bar{n})$ (Eq. (37)), while the dual variable $\mu$ is updated by dual ascent to enforce the mean rollout budget constraint (Eq. (38)). Operationally, $v_b(\theta)$ can be estimated from the same rollouts used to compute $\widehat{L}_b(\theta; n)$. For a prompt $x$ with $n$ rollouts $\{y_j\}_{j=1}^{n}$, let $g_j(x; \theta)$ denote the per-rollout GRPO policy-gradient contribution. A natural within-prompt proxy is the sample variance (biased vs. unbiased only changes constants)

$$\widehat{\mathrm{Var}}(g \mid x) \triangleq \frac{1}{n-1} \sum_{j=1}^{n} \left\| g_j(x; \theta) - \bar{g}(x; \theta) \right\|_2^2, \qquad \bar{g}(x; \theta) = \frac{1}{n} \sum_{j=1}^{n} g_j(x; \theta),$$

which can be aggregated over prompts in bin $b$ to form a bin-level proxy $\hat{v}_b(\theta)$. In practice we use a monotone scalar surrogate of this signal (e.g., the empirical variance of reward or advantage across rollouts within each bin) to guide arm selection, consistent with the variance-reduction viewpoint developed in Section E.2.1–Section E.2.3.

