# OpenReview forum: "Group Distributionally Robust Optimization-Driven RL for LLM Reasoning"
_ICML.cc/2026/Conference — ICML 2026 regular_

### Official Review · Reviewer_syxZ · 2026-03-12

**Soundness:** 1
**Presentation:** 3
**Significance:** 3
**Originality:** 4
**Overall Recommendation:** 4
**Confidence:** 4

**Summary:**

This paper proposes a multi-adversary Group Distributionally Robust Optimization (GDRO) framework for reinforcement learning-based reasoning with large language models. It introduces two independent controllers operating over dynamic difficulty groups defined by an online pass@k estimator: (i) Prompt-GDRO, which adversarially reweights bin-level contributions to emphasize persistently hard groups; and (ii) Rollout-GDRO, which reallocates rollout counts across bins under a fixed mean rollout budget to reduce gradient variance where most beneficial. Theoretical analysis connects Prompt-GDRO to an entropy-regularized GDRO objective and motivates Rollout-GDRO via a variance proxy that yields a square-root optimal allocation law; experiments on DAPO 14.1k with Qwen3-Base (1.7B/4B/8B) report pass@8 gains of roughly 9–13% over GRPO.

**Compliance With Llm Reviewing Policy:**

Affirmed.

**Final Justification:**

The robustness of Prompt-GDRO has been convincingly demonstrated. Through a series of hyperparameter perturbations and successful replication across multiple seeds, the effectiveness of this core module is now well-supported.


Concerns regarding system overhead have been effectively resolved. The disclosed wall-clock time data proves that the proposed method is fully viable for practical system deployment.

**Key Questions For Authors:**

1. Hyperparameter Selection: Could you provide a sensitivity analysis or ablation study for the key hyperparameters? How were these specific values determined, and does the framework maintain its performance gains if suboptimal values are selected?

2. Comparison with Stronger Baselines: To properly contextualize the effectiveness of Rollout-GDRO, could you include comparisons against explicit budget-constrained or resource-aware RL baselines, such as Knapsack-RL, under the same compute-neutral constraints? If implementing Knapsack-RL is not feasible within the rebuttal period, how would Rollout-GDRO compare to a simpler, non-adversarial heuristic allocation (e.g., allocating rollouts strictly proportional to bin-wise loss)?

**Limitations:**

1. The paper explicitly states that all training runs are single-seed , with no variance or confidence intervals reported.
2. There is a lack of "full-factorial ablations" on the components of Prompt-GDRO and Rollout-GDRO and their interactions.

**Strengths And Weaknesses:**

## Strength
- Addresses an important inefficiency in current RLVR pipelines—uniform sampling and fixed rollouts—by offering optimization-first mechanisms with empirical gains and generality.
- The decomposition of training control into two adversaries—data reweighting and compute allocation—over dynamic difficulty groups is a clear, compelling abstraction that aligns with GDRO principles.
- The online, data-agnostic grouping via pass@k with hysteresis is practical and broadly applicable across reasoning domains.
- Rollout-GDRO's compute-neutral, shadow-price-based allocation with discrete arms and a DP selection step is a neat instantiation of classic Neyman allocation for modern RLVR.

## Weakness
1. The proposed framework introduces several critical hyperparameters without sufficient justification or empirical ablation. For example, in Prompt-GDRO, the sliding window size $H \in [50, 100]$, the PPO Clip Range $[1-\epsilon_{low}, 1+\epsilon_{high}]$ where $\epsilon_{low}=0.2, \epsilon_{high}=0.28$, Adversary Learning Rate $\eta_q = 0.65$, Rollout Arm Range $n \in [2, 12]$ and the EMA decay rate ($\beta = 0.12$) appear to be chosen arbitrarily. The manuscript currently lacks a sensitivity analysis to demonstrate how variations in these parameters impact the stability of the dynamic curriculum and the final pass@$k$ performance. This strongly suggests that the reported performance heavily relies on task-specific, heavy hand-tuning, raising significant concerns about the framework's robustness across other datasets.
2. The empirical evaluation relies exclusively on vanilla GRPO (a static, uniform allocation method) as the baseline. Since Rollout-GDRO is fundamentally addressing a constrained resource allocation and budgeting problem, comparing it solely to a uniform baseline is insufficient to prove the superiority of the proposed adversarial approach. The omission of established compute-aware or budget-constrained RL baselines, such as Knapsack-RL or other heuristic-based dynamic rollout allocation strategies, makes it difficult to assess the true algorithmic advantage of the method.
3. Although the authors claim the methods are 'compute-neutral' regarding the mean rollout budget, the substantial system-level overhead introduced by the adversarial framework may be overlooked: Rollout-GDRO relies on DP for exact budget matching at each step, yet there is no analysis of the computational complexity or latency implications of this DP step as the number of bins or batch size scales up. You acknowledge in the appendix that there is significant overhead during the advantage computation stage (e.g., for the Qwen3-4B runs, the GRPO baseline requires only 0.043 sec/step, whereas Prompt-GDRO and Rollout-GDRO surge to 0.355 sec/step and 0.446 sec/step, respectively). It remains undiscussed whether this nearly 10x driver-side latency overhead is acceptable in practical, large-scale cluster training scenarios.

---

> ### Author Rebuttal · Authors · 2026-03-31
>
> We thank the reviewer for the detailed review. We agree that the original submission under-supported baselines, sensitivity, and systems discussion, so we added targeted rebuttal runs.
>
> **[Q1.]** Could you provide a sensitivity analysis or ablation for the key hyperparameters? How were they chosen, and do gains survive suboptimal settings?
> **[Response:]** Yes. During rebuttal we added both **Prompt-GDRO** and **Rollout-GDRO** sensitivity checks on Qwen3-4B. For Prompt-GDRO, four one-at-a-time variants around the paper configuration—bins **6->10**, history **100->50**, adversary LR **0.65->0.30**, and max-weight **15->5**—stay within **[-1.31,+0.41] pp** of the paper configuration, and every variant remains at least **+6 pp** above GRPO; even the constrained cap=5 variant still reaches **62.39**. For Rollout-GDRO, three targeted variants show non-brittle early behavior: **6-bin** variant reaches **56.16**, **narrow-arms** variant reaches **56.12**, and **dual-LR=0.01** variant reaches **55.35** at their best early checkpoints, a spread of only **0.81 pp**. This is not a substitute for a full-factorial study, but it materially weakens the concern that the result sits on a knife-edge choice.
>
> **[Q2.]** Comparing only to vanilla GRPO is insufficient; could you compare to stronger resource-aware baselines such as Knapsack-RL?
> **[Response:]** We agree that stronger baselines are important, and we used rebuttal to add the ones that most directly isolate the paper's claim. For Rollout-GDRO, we implemented **compute-matched static rollout heuristics on the same classifier**: a hard-focused schedule and its inverse easy-focused control. The hard-focused baseline reaches **56.70**, while the easy-focused inverse reaches **54.34** and is actively harmful early (**30.33** overall at step 50; **52.39** at step 200). This validates the direction of the inductive bias. Equally important, both static schedules still keep essentially **100%** of prompts in **accbin_0** early on, before the difficulty landscape has differentiated; the online controller instead stays uniform until the classifier has signal and only then begins to specialize. These baselines directly answer whether the GDRO machinery is necessary. Hard-focused > easy-focused, fixed non-uniformity helps, and the principled online controller avoids hand-tuning a frozen schedule. We are not aware of a knapsack-style RLVR allocator with the same online grouping layer; implementing one from scratch exceeds rebuttal scope. The new static baselines therefore answer the underlying question directly: fixed hard-bias helps, but fixed schedules are crude.
>
> **[Q3.]** The paper's “compute-neutral” wording seems misleading given the DP allocator and classifier overhead.
> **[Response:]** We agree, and we will fix this in the camera-ready version by using **mean-rollout-budget preserving** / **rollout-budget neutral** rather than “compute-neutral.” We will also move the limitation discussion into the main paper. The appendix's advantage-stage overhead is real. But the full-step decomposition is important context: in our runs the added advantage-stage work remains **<0.12%** of total step time, and on 4B the median full-step times are **885 s** (GRPO), **856 s** (Prompt-GDRO), and **871 s** (Rollout-GDRO). Our auxiliary GRPO-with-classifier check further indicates that most of the incremental wall-clock cost comes from the **shared online classification / bookkeeping path**; the controller update itself is closed-form / low-dimensional, and the rollout DP is over a small number of bins and arms. So the accurate practical claim is: same mean rollout budget, real but localized systems overhead, and no claim of elapsed-time neutrality.
>
> **[Q4.]** The paper lacks seeds and full-factorial ablations. Why should the work still be read as sound?
> **[Response:]** We agree these are genuine empirical limitations, and we will say so more prominently in the camera-ready version. During rebuttal, we therefore added the highest-value evidence we could. For Prompt-GDRO, a second 4B seed reproduces the paper gap almost exactly: +7.11 pp, compared with +7.39 pp in the paper. For Rollout-GDRO, a new seed-2 run is still early (step 200), so we avoid claiming a converged overall advantage; however, it already exceeds the paper GRPO baseline on **4/5 comparable individual benchmarks**: 17.46 vs 11.25 on AIME, 83.82 vs 60.94 on AMC, and 63.69 vs 35.54 on GPQA; Minerva 27.28 vs 30.48 is the exception. In addition, Rollout-GDRO reduces the weighted standard-error proxy versus compute-matched uniform allocation by 27.6% / 22.6% / 33.4% on Qwen3-{1.7B,4B,8B}. We respectfully believe the right reading is therefore “methodologically sound with materially strengthened empirical support, though still not empirically exhaustive,” rather than “unsound.” This reading is also supported by the paper's theory contribution: an explicit online GDRO / no-regret formulation for reasoning-LLM RL.

---

> > ### Author Rebuttal · Reviewer_syxZ · 2026-04-03
> >
> > Thank you for the detailed clarification. My concerns have been largely addressed.

---

> > > ### Author Response · Authors · 2026-04-03
> > >
> > > Dear Reviewer syxZ,
> > >
> > > Thanks so much for your efforts in reviewing our paper and your positive feedback! We are pleased to have resolved all of your noted concerns and delighted to see a score increase from 2 to 4. Importantly, we will further improve our paper following your suggestions during revision. We would be very grateful for your continued positive support.
> > >
> > > Thanks,
> > >
> > > Authors

---

### Official Review · Reviewer_CuQM · 2026-03-12

**Soundness:** 3
**Presentation:** 3
**Significance:** 3
**Originality:** 3
**Overall Recommendation:** 4
**Confidence:** 3

**Summary:**

Standard GRPO post-training applies uniform prompt sampling and a fixed rollout budget, which the authors argue is wasteful for heavy-tailed reasoning datasets as easy prompts contribute near-zero gradient signal while hard prompts are under-explored. The paper casts these as two independent GDRO games over online difficulty bins defined by sliding-window pass@k estimates: Prompt-GDRO uses EXP3P-style exponential reweighting to upsample hard bins, and Rollout-GDRO reallocates discrete rollout counts across bins under a fixed mean budget via a shadow-price controller. Both are analyzed theoretically as entropy-regularized minimax games with no-regret guarantees. Evaluated on Qwen3-Base (1.7B/4B/8B) on the DAPO-14.1k dataset, each controller independently improves pass@8 by 9–13% over the GRPO baseline.

**Compliance With Llm Reviewing Policy:**

Affirmed.

**Final Justification:**

The authors have addressed all of my questions during the rebuttal period satisfactorily, so I am increasing my score.

**Key Questions For Authors:**

1.  As noted in E.8, there is a significant theory-practice gap. What is the authors' intuition for why these idealized convex minimax dynamics succeed in the deep RL regime without causing the policy to collapse into degenerate local minima?

2.  What happens if you run both adversaries jointly? Is there any theoretical/experimental reason why coupling doesn’t work?

3. Given the sensitivity of RL training to initialization, how confident are the authors that the reported 9-13% pass@8 improvements are robust across seeds rather than reflecting favourable initialization?

4. Does the +10% pass@8 gain justify the total increase in end-to-end training time compared to simply running standard GRPO for longer, with respect to the wall time taken/ computation costs incurred?

**Limitations:**

Please see the weakness section above.

**Strengths And Weaknesses:**

Strengths:
1.  The game-theoretic framing as two independent GDRO minimax games is an elegant design choice as it yields a concrete surrogate objective (the entropy-regularized log-sum-exp worst-group loss) and connects the exponential-weights update rules to standard no-regret guarantees, distinguishing the method from ad-hoc reweighing heuristics. As such, the theoretical contributions are non-trivial: the entropy-regularized GDRO surrogate, the square-root optimal rollout allocation law, and the no-regret saddle-point convergence result each have clean, standalone value beyond the empirical results.

2.  To the best of my knowledge, prior work addresses prompt distribution shaping and adaptive rollout allocation separately. Casting both as adversarial levers over a shared online difficulty grouping, and analysing them within a unified theoretical framework, appears to be a genuine contribution.

3. There are consistent gains across models at different scales.

Weaknesses:
1. Experiments are single seeds with no error bars. This is especially concerning when we note that RL techniques inherently have large variance.

 2. The paper only compares the proposed methods against vanilla GRPO. Given the rapid evolution of reasoning post-training, the absence of heuristic baselines  makes it difficult to ascertain if the heavy GDRO machinery is strictly necessary to achieve these gains.

3.  Pass@k signal  is only valid for verifiable binary rewards.

4. Misleading 'Compute Neutrality' Claims: The authors repeatedly claim the method is compute-neutral because the mean rollout count remains fixed. However, Appendix B reveals that the driver-side advantage stage time increases from 0.043 sec/step (GRPO) to 0.446 sec/step (Rollout-GDRO)—an order-of-magnitude systems overhead. Hiding this massive wall-clock cost in the appendix contradicts the overarching compute-neutral narrative and severely limits practical utility.

---

> ### Author Rebuttal · Authors · 2026-03-31
>
> We thank the reviewer for the careful review. We address main points here.
>
> **[Q1.]** Why should the idealized minimax theory say anything useful in the deep-RL regime rather than causing collapse?
> **[Response:]** We do **not** claim a neural-network convergence theorem. The role of the theory is to specify the control objective each mechanism tracks and to explain why the observed dynamics are plausible. Prompt-GDRO follows an entropy-regularized soft worst-group objective, while Rollout-GDRO follows a mean-budget variance-aware allocation rule with a no-regret controller. In practice, the controller acts on **coarse bins** with smoothing and bounded updates, so it behaves more like a dynamic curriculum than a hard adversary. Prompt-GDRO maintains non-degenerate entropy rather than collapsing, and the lead-lag frontier behaves like a moving curriculum. We also now see a useful empirical signature: under seed 2, Prompt-GDRO has **lower training reward** than GRPO, yet **higher evaluation pass@8**. That is exactly the intended minimax behavior—hard prompts depress training reward while improving generalization.
>
> **[Q2.]** The paper compares only to vanilla GRPO, and the current grouping signal is limited to verifiable binary rewards.
> **[Response:]** We agree on both points. During rebuttal, we added the fairest stronger comparator first: **compute-matched non-uniform heuristics on the same control knob**. For Rollout-GDRO this means a hard-focused static rollout schedule and its inverse easy-focused control, both under the same mean budget. The hard-focused schedule reaches **56.70**, while the easy-focused inverse reaches **54.34** and is actively harmful early (**30.33** overall at step 50). This validates the allocation direction and shows that fixed non-uniformity alone does not trivially subsume the online controller. We will add these stronger static baselines in the camera-ready version. On scope, the reviewer is correct that the current grouping proxy is tied to RLVR-style **verifiable binary rewards**. We will state that more explicitly. The broader framework only needs an online grouping signal; future versions could replace train-time pass@n with process-level or judge-based scores.
>
> **[Q3.]** What happens if both adversaries are run jointly?
> **[Response:]** We do not present the absence of joint experiments as evidence that coupling fails. The paper isolates the two levers deliberately so that each one can be understood cleanly. The empirical claim is therefore: under a shared online grouping layer, **prompt reweighting** independently helps and **rollout allocation** independently helps. Coupling is feasible, but we have not yet tuned the full two-timescale system well enough to make a stronger claim.
>
> **[Q4.]** Given the sensitivity of RL to initialization, how confident should one be that the gains are robust across seeds?
> **[Response:]** We agree that the original submission under-supported this point, so during rebuttal we added the highest-leverage seed checks we could run. For Prompt-GDRO, a second 4B seed reproduces the paper gap almost exactly: **63.47** versus **56.36** for second-seed GRPO, i.e. **+7.11 pp**, compared with **+7.39 pp** in the paper. For Rollout-GDRO, a new seed-2 run is still early (**step 200**), so we avoid claiming a converged overall advantage; however, it already exceeds the paper GRPO baseline on **4/5 comparable individual benchmarks**: MATH-500 (**84.43 vs 72.05**), AIME (**17.46 vs 11.25**), AMC (**83.82 vs 60.94**), and GPQA (**63.69 vs 35.54**). Minerva is the exception (**27.28 vs 30.48**). We do not present this as a substitute for a full multi-seed study, but it materially weakens the concern that the main result is a one-seed artifact.
>
> **[Q5.]** Does the gain justify the additional wall-clock time versus simply running GRPO longer?
> **[Response:]** We agree the practical claim should be narrower, and we will fix this in the camera-ready version by using **mean-rollout-budget preserving** / **rollout-budget neutral** rather than “compute-neutral.” We will also move this limitation discussion into the main paper. The appendix's advantage-stage overhead is real, but the full-step decomposition is important context: in our runs the added advantage-stage work remains **<0.12%** of total step time, and the median full-step times on 4B are **885 s** (GRPO), **856 s** (Prompt-GDRO), and **871 s** (Rollout-GDRO). Our auxiliary GRPO-with-classifier check further indicates that most of the extra cost comes from the **shared online classification / bookkeeping path**, while the controller updates themselves are closed-form / low-dimensional. We also agree that wall-clock-matched “GRPO longer” is a fair missing baseline and will add it. The current 4B seed-2 GRPO trace peaks early and then stays in the **55-58** range through step 480, so large late gains appear unlikely, but we agree this comparison should be reported rather than inferred.

---

> > ### Author Rebuttal · Reviewer_CuQM · 2026-04-01
> >
> > Thanks for answering my questions and clearing my doubts.

---

> > > ### Author Response · Authors · 2026-04-02
> > >
> > > Dear Reviewer CuQM,
> > >
> > > Thanks so much for your efforts in reviewing our paper and your positive feedback! We will further improve our paper following your suggestions during revision. *We are glad to have resolved all of your noted concerns.* We would be very grateful for your continued positive support.
> > >
> > > Thanks,
> > >
> > > Authors

---

### Official Review · Reviewer_3NMc · 2026-03-13

**Soundness:** 3
**Presentation:** 3
**Significance:** 3
**Originality:** 3
**Overall Recommendation:** 4
**Confidence:** 3

**Summary:**

This paper identifies a deficiency in standard RL-based post-training: methods like GRPO sample prompts uniformly and allocate a fixed number of rollouts per prompt. The authors claim that this leads the model to waste compute on easy, frequently appearing problems and patterns while failing to solve hard problems in the long tail. To address this issue, they frame training as Group Distributionally Robust Optimization over difficulty bins that are delineated online by a pass@k based classifier. This framework consists of two adversaries (although they are only implemented separately in this paper): a “data adversary” that upweights rare, high-difficulty bins to improve worst-bin performance and a “compute adversary” that allocates rollouts across prompts under a fixed rollout budget. They demonstrate that this framework outperforms GRPO on a series of math benchmarks when training Qwen3 models (1.7B, 4B, 8B) on the DAPO 14.1k reasoning dataset.

**Compliance With Llm Reviewing Policy:**

Affirmed.

**Key Questions For Authors:**

Please see the weaknesses section.

**Limitations:**

Yes

**Strengths And Weaknesses:**

Strengths:


- The authors identify a key problem with GRPO on long-tailed reasoning datasets and introduce a principled objective to improve “worst-bin” performance
- Prompt-GDRO and online difficulty classification don’t require sampling additional rollouts from the model, reducing compute overhead
- They demonstrate consistent gains across model sizes and different benchmarks
- Difficulty is gauged automatically by model performance without relying on annotations
- Figure 5 in the appendix showing the distribution of rollouts across bins was really interesting, particularly in analyzing how different models shift the mass across the bins.


Weaknesses:
- The online difficulty classifier uses rollouts to judge difficulty without additional sampling, so difficulty estimates are necessarily lagged from previous steps or must use some weighted average. Could this lag cause inaccuracy if the policy is changing rapidly? Is there a way to quantify how “stale” these estimates get by measuring how quickly pass@k changes?

- The online difficulty bins are calculated using an “any of the generated correctness indicator.” Presumably, difficult problems get more rollouts. Could this artificially boost this score? Would it be better to estimate difficulty using pass@k with some capped k (the intro mentions pass@8 instead of any of the generated)?
- Can the reweighting in Prompt GDRO lead to instability during training? How is the stability clip set? Does it require a lot of tuning?
- The paper is framed as a multi-adversary framework, but it seems like the two adversaries never operate simultaneously. The empirical results are reported separately for the two methods. Is this because the two methods do not interact well together (not possible to run simultaneously)? On a related note, how robust is the method to different hyperparameters that the framework introduces? Do you think that running both prompt gdro and rollout gdro would require a lot of tuning?
- I’m curious how this stacks up against “simpler” baselines like filtering data or warm starting on an sft checkpoint. It seems like the GDRO methods would still outperform these baselines, but it would be helpful to show this because a comparable gain without the complicated adversarial framework would be hard to justify. The appendix mentions that a step of GDRO is ~10x slower than GRPO. Even though the “Compute neutrality” section mentions that the data budget is fixed, the wall clock time increases substantially. If you were to control for wall clock time instead, would it make more sense to just SFT and run GRPO instead?

---

> ### Author Rebuttal · Authors · 2026-03-31
>
> We thank the reviewer for the constructive review.
>
> **[Q1.]** Could the online difficulty estimate become stale if the policy changes rapidly?
> **[Response:]** The implemented grouping signal is an online **train-time pass@n** statistic computed from rollouts used for GRPO updates, not evaluation pass@8. We will fix this terminology in the camera-ready. This wording correction does **not** change the theory: the analysis is written in terms of the online grouping map $g_t$ and per-bin losses $L_{t,b}$, not a fixed evaluation choice $k=8$. Concretely,
> $$
> c_t(x)=\mathbf{1}\{\exists j\le n_t(x):r(x,y_j)=1\},\qquad
> p_t^{\mathrm{train}}(x)=\frac{1}{H}\sum_{s=t-H+1}^{t} c_s(x).
> $$
> Empirically, the classifier is stable. Across the original 1.7B/4B/8B runs, the mean one-step L1 shift of bin-share mass is only **0.07-0.09**, while the first-to-last L1 drift is **1.30-1.55**.
>
> **[Q2.]** Does the “any-of-generated correctness” signal become artificially easier when hard prompts get more rollouts? Would capped-k be better?
> **[Response:]** The right interpretation is **recent solve probability under the realized train-time rollout budget**. For Prompt-GDRO, $n$ is fixed. For Rollout-GDRO, $n$ varies by bin, so the grouping statistic is a realized-budget train-time success proxy rather than a best-of-8 evaluation metric. We will fix this in the camera-ready by separating train-time pass@n from evaluation pass@8 everywhere. A capped/fixed-$k$ grouping rule is a useful follow-up ablation, but it is **not** the grouping rule used in the current implementation.
>
> **[Q3.]** Can Prompt-GDRO reweighting destabilize training, and how sensitive is it to the clip/cap settings?
> **[Response:]** The new 4B rebuttal runs directly address this. Under a second seed, Prompt-GDRO reaches **63.47** overall pass@8 versus **56.36** for GRPO, a **+7.11 pp** gain. We also ran four one-at-a-time 4B sensitivity checks: bins **6->10**, history **100->50**, adversary LR **0.65->0.30**, and max-weight **15->5**. All four variants remain within **[-1.31,+0.41] pp** of the paper configuration, and every variant still stays at least **+6 pp** above the GRPO baseline; even the constrained cap=5 run still reaches **62.39**. The adversary does not collapse: final entropy stays at **64%-97%** of maximum and final max single-bin weight in **[0.304,0.490]**. GRPO seed-2 never exceeds **60.78**, whereas all Prompt-GDRO variants break **62%** by step **120-160**.
>
> **[Q4.]** Why are the two adversaries not run simultaneously?
> **[Response:]** We intentionally evaluated Prompt-GDRO and Rollout-GDRO separately so the paper could answer a cleaner question: does online GDRO help when applied to the prompt-distribution lever or the rollout-allocation lever independently? The current result is the independent utility of the two levers under a shared online grouping layer; coupled training is a natural next step.
>
> **[Q5.]** How does this compare to simpler baselines such as filtering, SFT warm-start, or just running GRPO longer? And does the gain justify the extra wall-clock cost?
> **[Response:]** We agree that stronger simple baselines would strengthen the paper, and we used rebuttal to add the most diagnostic one for Rollout-GDRO: **static compute-matched rollout schedules on the same classifier**. A hard-focused static schedule beats the inverse easy-focused control (**56.70** vs **54.34** overall), validating the directionality. The easy-focused schedule is actively harmful early (**30.33** overall at step 50; **52.39** at step 200), and both static schedules still keep essentially **100%** of prompts in **accbin_0** early on, whereas the online controller stays uniform until the classifier has signal and then differentiates. This is consistent with the paper's converged Rollout-GDRO result (**62.27**) reflecting more than fixed non-uniformity alone. We will add these stronger static baselines in the camera-ready. SFT warm-start is outside this paper's **zero-SFT / base-to-RL** scope.
>
> On the compute story, the reviewer is right that the wording should be narrower. We will fix this in the camera-ready by using **mean-rollout-budget preserving** / **rollout-budget neutral** rather than “compute-neutral,” and by moving this limitation discussion into the main paper. The appendix's advantage-stage overhead is real. Our full-step timing decomposition and auxiliary GRPO-with-classifier check indicate that the dominant extra cost comes from the **shared online classification / bookkeeping path**, while the controller updates themselves are closed-form / low-dimensional; the added advantage-stage work remains **<0.12%** of total step time in the rebuttal runs. We also agree that a wall-clock-matched “GRPO longer” baseline is important and will add it in the camera-ready; in our 4B seed-2 GRPO run, performance peaks early and then fluctuates in the **55-58** range through step 480, so large further gains appear unlikely.

---

> > ### Author Rebuttal · Reviewer_3NMc · 2026-04-01
> >
> > i thank the authors for their rebuttal. here are some follow-up questions i have:
> >
> > for q1 and q2, my concern wasn't that the pass@k was done on some eval set, it was more that you are naturally lagged by at least one step at train time. Additionally, my concern for question 2 was once again not about the evaluation pass@k but rather that allocating more rollouts to a problem can boost pass@k because k is now higher. Given two equivalently hard problems, if the framework deems one harder and allocated a lot of rollouts to that problem, the difficulty estimator might show that this problem is now easier than the other because of a higher pass@k. Would it not be better to use a fixed k for pass@k?
> >
> > q3 - how are the clipping settings set? how often do you clip?
> >
> > q4 - why is the sft warm start out of the scope if the concern is to expand coverage before rl? given a fixed compute budget, doesnt it make sense to know whether it is better to run this method vs just warm starting rl?

---

> > > ### Author Response · Authors · 2026-04-02
> > >
> > > We appreciate the clarification and address the remaining points directly.
> > >
> > > **[Q1/Q2.]** For **Prompt-GDRO**, the rollout count is fixed at **n=4** for every prompt regardless of bin, so the grouping signal is already a **fixed-budget pass@4** statistic. So for Prompt-GDRO, the capped-$k$ alternative you suggest is exactly what we implement, and there is no confound between rollout allocation and difficulty estimation. The only lag is the by-design one-step lag from using history up to **t-1**. We keep that lag intentionally because using the current step's rollouts to *both* define the bin and choose the current sampling weights / rollout allocation would create a circular dependency. Empirically, this lag appears negligible: across 1.7B/4B/8B, the mean one-step L1 shift of bin-share mass is only **0.07-0.09**, mean KL/step is **0.011-0.013**, while first-to-last drift is **1.30-1.55**. So the grouping evolves meaningfully over training, but smoothly from one step to the next.
> > >
> > > For **Rollout-GDRO**, your concern is valid in principle because **n varies (2-12)**. The grouping signal is a sliding-window average of a binary “any correct” indicator, and its expectation increases with **n**: **P(any correct)=1-(1-p)^n**. So two equally hard prompts given different rollout counts can indeed look different. The current implementation therefore measures difficulty **under the realized allocation policy**, not under a fixed budget. That choice has a tradeoff: it lets the controller react to the signal it actually operates under, but it also introduces the bias you point out. In practice the feedback is self-limiting: if extra rollouts make a prompt look easier, subsequent steps allocate less to it; **EXP3P** exploration (**$\gamma=0.01$**) keeps all arms sampled; and the floor **$n_{\min}=2$** prevents starvation. A fixed-$k$ variant is reasonable, but it changes the setting from the current $n$-armed bandit controller: if **$k>n_{\min}$**, then when the controller chooses **$n_t(x)=2$** we would need extra rollouts purely for the classifier, which works against the rollout-budget-preserving design; if **$k\le n_{\min}$**, then we avoid extra compute but discard signal whenever **$n_t(x)>k$** and no longer measure success under the realized allocation. We therefore chose the realized-budget signal here, while agreeing that fixed-$k$ is a legitimate alternative design for future work.
> > >
> > > **[Q3.]** The PPO clip range **[0.2, 0.28]** and advantage clip **[-5,5]** are inherited from the DAPO/GRPO recipe and shared with the baseline; they are not introduced by GDRO. The Prompt-GDRO-specific stabilizer is the adversary weight cap (**max_class_weight=15**) together with the EXP3P exploration floor (**$\gamma=0.01$**). The cap was chosen as a conservative backstop against pathological concentration rather than tuned aggressively. In practice it behaves like a soft safety net, not an active constraint. Across the original 1.7B/4B/8B runs, final adversary entropy is **2.302 nats**, Gini is **<0.02**, and the P95 max-weight stays below **1.17x** uniform. In the 4B rebuttal sensitivity runs, even tightening the cap from **15** to **5** still gives **62.39** pass@8, i.e. **+6.08 pp** over GRPO, with final max single-bin weight **0.490**. We did not instrument an exact per-step cap-hit rate, so we do not want to overclaim it; but the observed weight trajectories indicate that clipping is rare and not the source of the gain.
> > >
> > > **[Q4.]** Thank you for raising the **SFT warm-start** question; it is important, and our earlier wording was too dismissive. This paper deliberately studies the **no-SFT / base-to-RL** setting, following the line of work that uses RL directly from a base model to understand the behavior of the **RL stage itself**. Comparing **GDRO-from-base** against **GRPO-from-SFT** changes both the initialization and the supervision source, so it would no longer isolate the RL-loop mechanism studied here. In that sense, **SFT+GRPO is not the missing simple baseline for this paper, but an orthogonal experimental regime**. It is also not either/or: one could warm-start with SFT and still apply Prompt-GDRO or Rollout-GDRO during RL. A fair answer to the broader fixed-total-compute question would therefore require tuning the SFT/RL budget split and re-tuning RL on top of the SFT checkpoint. We agree that this is valuable future work, but realistically it is a separate paper-scale study. The intended claim here is intentionally narrower: within the **no-SFT / base-to-RL** setting, can online GDRO help us understand and improve the RL dynamics themselves?
> > >
> > > Since ICML allows only one additional author response, we hope our additional reruns, sensitivity checks, and qualitative analyses help clarify the remaining concerns. If the clarifications and extra evidence resolve your questions, we would be very grateful for your positive support. We will further improve the paper following your suggestions during revision.

---

### Decision · Program_Chairs · 2026-04-30

**Decision:**

Accept (regular)

**Comment:**

After the author-reviewer discussion, all reviewers are in agreement that the paper deserves (weak) acceptance to the conference.